# Generating high-order optical and spin harmonics from ferromagnetic monolayers

G.P. Zhang [1], M.S. Si[2], M. Murakami[1], Y.H. Bai[3] & Thomas F. George[4]

High-order harmonic generation (HHG) in solids has entered a new phase of intensive research, with envisioned band-structure mapping on an ultrashort time scale. This partly benefits from a flurry of new HHG materials discovered, but so far has missed an important group. HHG in magnetic materials should have profound impact on future magnetic storage technology advances. Here we introduce and demonstrate HHG in ferromagnetic monolayers. We find that HHG carries spin information and sensitively depends on the relativistic spin–orbit coupling; and if they are dispersed into the crystal momentum **k** space, harmonics originating from real transitions can be **k**-resolved and carry the band structure information. Geometrically, the HHG signal is sensitive to spatial orientations of monolayers. Different from the optical counterpart, the spin HHG, though probably weak, only appears at even orders, a consequence of SU(2) symmetry. Our findings open an unexplored frontier—magneto-high-order harmonic generation.

[1] Department of Physics, Indiana State University, Terre Haute, IN 47809, USA. [2] Key Lab for Magnetism and Magnetic Materials of the Ministry of Education, Lanzhou University, Lanzhou 730000, China. [3] Office of Information Technology, Indiana State University, Terre Haute, IN 47809, USA. [4] Office of the Chancellor Departments of Chemistry & Biochemistry, Physics & Astronomy University of Missouri-St. Louis, St. Louis, MO 63121, USA. Correspondence and requests for materials should be addressed to G.P.Z. (email: gpzhang.physics@gmail.com)

High-order harmonic generation (HHG) in atoms and small molecules has garnered attentions over several decades. It allows one to generate a table-top light source with energy up to x-ray regimes and time scales down to several hundred attoseconds[1–3]. This leads to the advent of attosecond physics[4], where electron dynamics is probed on its intrinsic time scale[5]. Farkas and coworkers[6] were the first to generate 5th-order harmonics from a gold surface by a picosecond laser pulse. von der Linde et al.[7] reported up to 15th order in an Al film and 14th order in glass. Theoretically Plaja and Roso-Franco[8] examined the mechanism of harmonic generation in silicon, while Faisal et al.[9] developed a nonperturbative Floquet–Bloch theory to control HHG through interband resonances. In 2005, we predicted HHG in $C_{60}$ theoretically[10,11] (see other references cited there), and Ganeev et al.[12,13] experimentally demonstrated HHG in fullerenes. However, nanostructures[14] are traditionally unfamiliar to researchers in atomic HHG[15]. In 2011, HHG in ZnO reported by Ghimire et al.[16] renewed the interest in solid state HHG, which has quickly expanded into monolayer[17] and multilayer graphene[18], MgO[19,20], Si[21], MoS$_2$[22], Bi$_2$Se$_3$[23], SiO$_2$[24,25], Ar/Kr solids[26], GaSe[27–29], and metal-sapphire nanostructures[30]. However, to this end, little attention has been paid to magnetic systems[31,32].

Here we show that a single laser pulse is capable of generating HHG in iron monolayers. Different from nonmagnetic materials, the harmonic signal carries the spin information. The majority and minority spins generate different harmonics. In contrast to HHG in atoms and small molecules, circularly polarized light can generate even higher-order harmonics, which are helicity-dependent. We compare two different Fe(110) and Fe(001) surfaces and find that the harmonics from Fe(110) are stronger. We find that the different density of states around the Fermi level is responsible for this difference. We disperse harmonics in the crystal momentum space, and we find that, in general, harmonics, which result from virtual transitions, appear symmetric with respect to the harmonic order and carry no information on the band structure. However, if harmonics originate from real transitions, they can be attributed to a few specific transitions between band states and are useful for band structure reconstruction. Higher harmonics show a stronger band dispersion. Different from the charge counterpart, due to the SU(2) symmetry, the harmonics from spin appear at even orders. Our study opens a new direction by extending high-harmonic generation to magnetic materials.

## Results

### Symmetry properties of magneto-high-order harmonic generation (MHHG).

HHG in nonmagnetic materials is only subject to the spatial symmetry. MHHG in magnetic materials is very different, where spin polarization and spin–orbit coupling break the spatial symmetry and introduce new phenomena that are otherwise undetectable. This already occurs in magneto-optics, such as the Faraday or Kerr effect. For instance, a cubic structure, if its magnetization is placed along the $z$ axis, becomes a tetragonal structure, and the number of symmetry operations is reduced from 48 to 16. Different from polar vectors, magnetic moment vectors $\mathbf{M}$ are axial vectors and transform as

$$T_{\text{axial}}(O)\mathbf{M} = \det[O]O\mathbf{M}, \qquad (1)$$

where det is the determinant of the symmetry operation $O$ and $T$ is the transformation. There is no difference between the polar and axial vectors for proper rotations, but for improper rotations, they differ by a sign change determined by the above determinant.

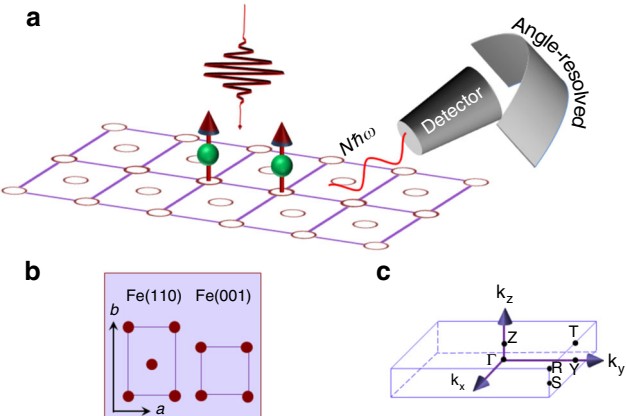

**Fig. 1** High-order harmonic generation in ferromagnetic materials. **a** An intense laser pulse excites a ferromagnetic iron monolayer and generates high-order harmonics. The harmonic has spin signature on the spectrum and can be dispersed in the crystal momentum space, so the harmonic peak can be assigned to a unique transition between occupied and unoccupied states. Higher-order harmonics are more sensitive to the band structure change than lower-order ones, thus making HHG an ideal tool for spin-resolved detection. **b** Brillouin zone of a simple orthorhombic structure. **c** Two film orientations—Fe(110) and Fe(001)—are placed in the $ab$ (or $xy$) plane

For an orthorhombic system with the magnetization along the $z$ axis (Fig. 1a), there are eight symmetry operations (see Methods), but only four of them keep the magnetization invariant and are retained in the symmetry group. These symmetry operations are essential to our understanding of MHHG. Consider a proper rotation $O_2$ (a twofold rotation around the $z$ axis, $C_{2z}$) and an improper rotation $O_6$ (a reflection with respect to the $yz$ plane, $\sigma_x$):

$$O_2 = \begin{pmatrix} -1 & 0 & 0 \\ 0 & -1 & 0 \\ 0 & 0 & 1 \end{pmatrix}; \quad O_6 = \begin{pmatrix} -1 & 0 & 0 \\ 0 & 1 & 0 \\ 0 & 0 & 1 \end{pmatrix}. \qquad (2)$$

For a nonmagnetic system, both symmetry operations appear in the point group. If the laser field is polarized along the $x$ axis, these two symmetry operations cancel any harmonic signal along the $y$ axis. Now consider the same laser field incident on a magnetic sample (see Fig. 1). If the system is spin-polarized along the $z$ axis, the point group only retains $O_2$, not $O_6$ since $O_6$ changes the direction of the spin moment. In other words, this symmetry reduction voids the original cancellation, so a new signal appears along the $y$ axis. On the other hand, it is easy to verify that if the laser polarization is along the $z$ axis, there is no signal along other directions. The entire symmetry properties can be worked out once the symmetry group is known. This is the theoretical foundation of MHHG.

### First-principles formalism.

We choose iron monolayers as our model system. Figure 1b shows two spatial orientations in the iron monolayers—Fe(110) and Fe(001) surfaces. They are simulated by a slab geometry, where a vacuum spacing separates slabs so there is little interaction between them (for details, see the Supplementary Methods). We solve the Kohn–Sham equation (in atomic units)[33] to find both the eigenstates and optical transition matrices,

$$\left[-\nabla^2 + V_{\text{ne}} + V_{\text{ee}} + V_{\text{xc}}\right]\psi_{i\mathbf{k}}(r) = E_{i\mathbf{k}}\psi_{i\mathbf{k}}(r), \qquad (3)$$

where the terms on the left-hand side represent the kinetic energy, nuclear–electron attraction, electron–electron Coulomb repulsion and exchange correlation[34], respectively. $\psi_{i\mathbf{k}}(r)$ is the Bloch wavefunction of band $i$ at crystal momentum $\mathbf{k}$, and $E_{i\mathbf{k}}$ is the band energy. We include the spin–orbit coupling using a second-variational method in the same self-consistent iteration[35] and construct the spin matrices. Once our calculation reaches self-consistency, we investigate harmonic generations by employing the time-dependent Liouville equation,

$$i\hbar\langle i\mathbf{k}|\frac{\partial\rho}{\partial t}|j\mathbf{k}\rangle = \langle i\mathbf{k}|[H_0 + H_I, \rho]|j\mathbf{k}\rangle, \quad (4)$$

where $\rho$ is the density matrix, $H_0$ is the system Hamiltonian, and $H_I$ is the interaction between the system and laser field: $H_I = \frac{e}{m_e}\widehat{\mathbf{P}} \cdot \mathbf{A}(t)$, where $-e$ is the electron charge, $m_e$ is its mass, $\widehat{\mathbf{P}}$ is the momentum operator and $\mathbf{A}(t)$ is the laser-field vector potential (see Supplementary Information). We choose a Gaussian pulse with duration $\tau$ and photon energy $\hbar\omega$. We note that the time-dependent Liouville density functional theory[36] is advantageous since it rigorously respects the Pauli exclusion principle that two electrons cannot occupy the same spin state at the same time.

After we numerically solve the density matrix $\rho$ from Eq. (4), we can compute the expectation value of the momentum operator[10,11] by $\mathbf{P}(t) = \sum_{\mathbf{k}} \text{Tr}\left[\rho_{\mathbf{k}}(t)\widehat{\mathbf{P}}_{\mathbf{k}}\right]$, where the trace is over band indices and includes interband contributions. We only include intraband transitions indirectly through the interband transition. For our current laser field amplitude, the crystal momentum shift is very small. The harmonic spectrum is computed by Fourier transforming $\mathbf{P}(t)$ into the frequency

domain through[10,37],

$$\mathbf{P}(\Omega) = \int_{-\infty}^{\infty} \mathbf{P}(t)e^{i\Omega t}\mathcal{W}(t)\mathrm{d}t, \quad (5)$$

where $\mathcal{W}(t)$ is the window function (see Supplementary Note 1). All the harmonic spectra below use $\log_{10}|\mathbf{P}(\Omega)|$. Caution must be taken that the time propagation during simulation must be long enough to resolve fine structures in HHG spectra. In our calculation, the starting time is $-600$ fs, and the ending time is typically around 600 fs and in some cases up to 1.5 ps. The time step, which determines the highest harmonic order, is 1/32 the laser period, and when the field is stronger, it is 1/64 the laser period. Both the extremely long time propagation and small time step ensure that our spectrum is very sharp and has a well-defined shape.

**Spin-polarized HHG.** We employ a 60-fs linearly polarized laser pulse along the $y$ axis. Our photon energy is $\hbar\omega = 2$ eV and field amplitude is 0.09 V/Å (far below Bragg reflection[27]). These parameters are commonly used in experiments and are employed for the following calculations. We start with a nonmagnetic Fe (110) monolayer where we run a nonspin-polarized calculation. Figure 2a shows that the harmonic signals are only along the $y$ axis, consistent with the above symmetry analysis, and the highest harmonic order is 13. The top curve is the one obtained with $\mathcal{W}(t) = 1$ and the bottom one with a hyper Gaussian function. Next, we consider a spin-polarized case without spin–orbit coupling. Ferromagnetic materials have two distinctive spin channels: majority (spin-up) and minority (spin-down). We carry out two separate calculations with the same laser parameters. Figure 2b shows the results for the majority spin, where for the same symmetry reason, no signal is found along either the $x$ or $z$ axis. The HHG signal for the spin-up channel also reaches up to 13th

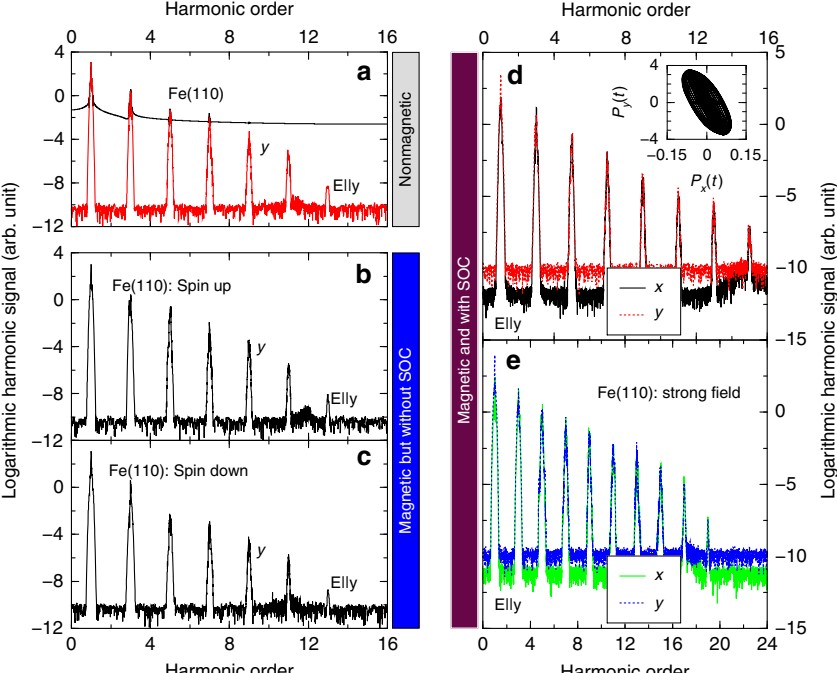

**Fig. 2** Harmonic signals under different magnetic ordering and laser amplitudes. **a** HHG signal from a nonmagnetic Fe monolayer. The laser E-field is along the $y$ axis, with $\hbar\omega = 2.0$ eV, $\tau = 60$ fs and field amplitude $E_0 = 0.09$ V/Å, for the results in this figure. The top curve is obtained without using the window function, while the bottom is processed with the window function. **b** HHG from the spin-up channel in a magnetic Fe monolayer. The spin–orbit coupling is not included. **c** Similar to **b**, but from the spin-down channel. **d** HHG signal with spin-polarized electrons and spin–orbit coupling. The solid and dashed lines denote the signals along the $x$ and $y$ axes, respectively. The spin is orientated perpendicular to the Fe(110) surface. Inset: Phase diagram of $P_x(t)$ versus $P_y(t)$. **e** Same as **d** but with $E_0 = 0.15$ V/Å, where high harmonics up to 19th order are observed

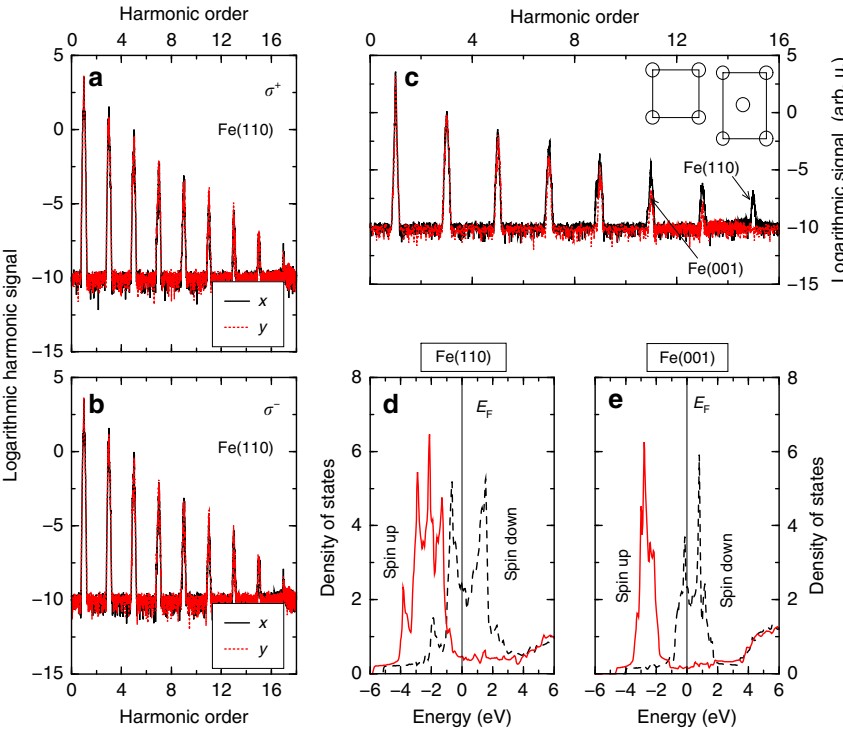

**Fig. 3** Effects of laser-helicity and film orientation on harmonic signals. **a**, **b** Logarithmic harmonic signal from the Fe(110) monolayer with right ($\sigma^+$) and left ($\sigma^-$) circularly polarized light, respectively, where the laser polarization is in the $xy$ plane (see Fig. 1). Due to the window function, the difference between the $x$ and $y$ components is not obvious, but the real time $P_y(t)$ is larger than $P_x(t)$ for most of the time (see Supplementary Fig. 3). **c** Comparison between HHG signals in the Fe(110) and Fe(001) monolayers, where the laser polarization is along the $z$ axis. **d** Density of states for the Fe(110) monolayer. The solid and dashed lines denote the spin-up and spin-down density of states, respectively. The Fermi level is at $E_F = 0$ eV (see the thin vertical line). **e** Density of states for the Fe(001) monolayer

order. However, the spin-down channel is quite different. Figure 2c shows that although its highest order is the same, the magnitude is smaller. We will come back to this below. What is even more interesting is when the spin–orbit coupling (SOC) is present. In this case, two spin channels are coupled, and the spin has a preferred spatial orientation, which breaks the symmetry. Figure 2d shows that this symmetry breaking introduces a new signal along the $x$ axis (see the solid line), along with the ordinary harmonics along the $y$ axis. This is qualitatively different from the nonmagnetic case where no signal is found along the $x$ axis (see Fig. 2a). The inset shows the real time evolution of $P_x(t)$ and $P_y(t)$. It is very interesting that similar to the time-resolved magneto-optical Kerr effect (TRMOKE)[38], these two components have a clear phase relation, and the major axis of the ellipse formed by $P_x(t)$ and $P_y(t)$ tilts slightly away from the $y$ axis. In TRMOKE, the angle that the major axis makes with the $x$ axis sensitively reflects the strength of the spin–orbit coupling. A similar feature observed here will be investigated in the future. To be sure that the HHG signal is indeed from the laser field, we increase the field amplitude to 0.15 V/Å, and find that the harmonic order increases all the way up to 19th order (see Fig. 2e). This demonstrates that our results are robust. If we compare those high harmonics with the low harmonics, we find that their signals do not drop significantly, so they should be measurable. Experimentally, the second-order harmonic was already observed[39]. This constitutes our first major finding that HHG in magnetic materials is spin-channel dependent and is affected by spin–orbit coupling, a unique feature that is not shared with other materials.

**Helicity and surface orientation dependence.** Different from HHG in atoms[1], we find that circularly polarized light[20] can effectively generate HHG in magnetic systems as well. We choose

right ($\sigma^+$) and left ($\sigma^-$) circularly polarized light in the $ab$ plane (see Fig. 1). In traditional magneto-optics, because of the spin–orbit coupling and exchange interaction[38], $\sigma^+$ and $\sigma^-$ do not generate identical signals because they choose different sets of transitions among band states. Figure 3a, b shows that HHG retains this difference, consistent with the prior magnetic second-order harmonic generation[40]. Energetically, our highest harmonic energy is significantly higher than that in native graphene[17,18].

There are additional knobs that one can turn in monolayers. They can be cut along different axes, so that the crystal orientation plays a role[28]. Fe monolayers have two possible orientations, Fe(110) and Fe(001) surfaces. Figure 3c shows that HHG in the Fe(110) monolayer is stronger than that in the Fe(001) monolayer; and this remains true even with a denser $k$ mesh (see Supplementary Information). Such orientational dependence has been reported before in GaSe[28] and $a$-cut (11–20) ZnO[41], but never in a magnet. You et al.[20] explained the orientation dependence in insulating MgO through the interatomic bonding. We do it differently, as electrons in our system are very delocalized.

One obvious explanation is that the number of atoms in the primitive cell is different for Fe(110) and Fe(001), but this is not the entire story. We notice that the harmonic amplitude ratio between Fe(110) and Fe(001) is not proportional to the atom number ratio. For instance, the ratio in the $z$ component is 2.33 for the first, 2.93 for the third, 3.34 for the 5th, 12.14 for the 7th, 38.01 for the 9th, and 28.05 for the 11th order. There is no signal at the 15th harmonic for Fe(001). We decide to examine the density of states (DOS) for the majority and minority bands. Figure 3d shows the total DOS in Fe(110) for the majority and minority states around the Fermi level $E_F$ (vertical thin line). The majority channel has more electrons than those in the minority

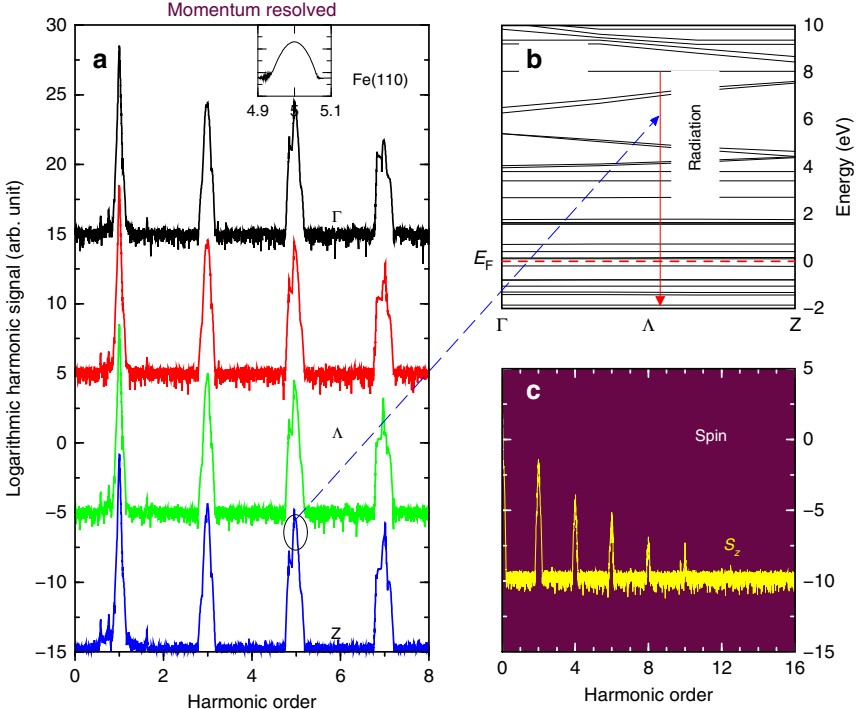

**Fig. 4** Crystal-momentum-resolved high-harmonic generation and spin harmonic generation. **a** Crystal-momentum-resolved harmonic signal from the $Z$ to $\Gamma$ point on a logarithmic scale. The laser is linearly polarized along the $y$ axis, and the signal is from the $x$ axis. Except the one at $Z$, all the curves are shifted vertically. Top inset: Zoomed-in view of the 5th harmonic at a different $k$ point. **b** Band dispersion along the $\Gamma$–$Z$ direction for the Fe(110) monolayer. The 5th harmonic corresponds to several crucial transitions from conduction states between 8 and 9.36 eV above the Fermi level (see the horizontal dashed line) to a state at $-1.87$ eV below the Fermi level. The arrow denotes this radiation. **c** Spin harmonic generation spectrum. Its zeroth order denotes the demagnetization, while higher orders represent the oscillation. The harmonic only appears at even orders, due to SU(2). The laser field has to interact with the system an even number of times to affect spins. These signals are normally weaker than the emission from the electric dipole

channel, so it contributes more to harmonic generation. This explains the difference seen in Fig. 2b, c. We see that due to the low symmetry in Fe(110), the majority valence electrons have three broad peaks, and the minority ones also have a large DOS below $E_F$. By contrast, the Fe(001) monolayer is quite different. Figure 3e shows that its majority band is much narrower than that in the Fe(110) monolayer, centered around 2.8 eV below the Fermi level. The band is less dispersive than for the Fe(110) monolayer, so many channels are not available to the Fe(001) monolayer to generate harmonics, which weakens HHG in the Fe (001) monolayer (see Supplementary Information for more).

## Discussion

Such a sensitive dependence of harmonic signals on DOS found here is important, as it suggests a potentially useful application to map band states through HHG in magnetic materials. This reminds us of our earlier work on $C_{60}$[10], where nearly every harmonic can be uniquely assigned to a particular transition. In ZnO, Vampa et al.[42] proposed to reconstruct the band structure from the HHG spectrum. Experimentally, Luu et al.[24] showed that the EUV radiation allowed them to probe the conduction band dispersion in SiO₂. A similar result was found in rare-gas solids[26]. But none of these studies tested magnetic materials. In the following, we demonstrate a highly accurate yet challenging detection scheme that the band transition states can be probed through HHG in the crystal momentum space. We take Fe(110) as an example. We disperse the harmonic signal in the crystal momentum space. There are many possible pockets in the Brillouin zone that we can investigate. Here we choose the $\Lambda$ line that links the $\Gamma$ and $Z$ points (see Fig. 1c). The harmonic signal is dispersed along the $\Gamma$–$Z$ direction, $\Lambda$ line from top to bottom in

Fig. 4a. Note that in our calculation all $\Gamma$ and $Z$ points are approximate since we have to shift the $k$ mesh slightly in order to get better convergence. For clarity, in Fig. 4a we vertically shift all the curves except the bottom one. These spectra carry rich information about the band states. We see that harmonics at different orders change with $k$ differently and this change is not limited to the lower-order harmonics. Higher-order harmonics show an even stronger dispersion. We take the fifth harmonic as an example. We see there are many smaller peaks. These peaks do not distribute symmetrically around the nominal 5th order. This is an important indication that the harmonic engages real transitions among band states[10], where the harmonic carries the band structure information and thus allows one to crystal-momentum resolve the bands.

Through a tedious but straightforward procedure (see details in the Supplementary Note 2), we pinpoint the origin of the 5th harmonic. It comes from radiation (see the arrow in Fig. 4b) from five conduction-band states between 8.0 and 9.36 eV above $E_F$ to a valence state around $-1.87$ eV below the Fermi level $E_F$ (dashed line). Therefore, interband transitions dominate the spectrum. We verify that if we remove these transition states, the 5th harmonic reduces sharply. The largest transition matrix element for the 5th harmonic is $(-0.4 - i0.068)10^{-2}\hbar/$bohr. To develop a generic picture of the limits of the crystal-momentum-resolved HHG, we generate a different set of $k$ points and compute their HHG spectra. We show one example in the top inset in Fig. 4a, where we see a nice symmetric Gaussian-like distribution around the 5th order. We find that these symmetric peaks normally result from virtual excitations, carry no information about the band states, and cannot be resolved in the crystal momentum space. Even if we

systematically exclude relevant transitions, we cannot cleanly remove the peak until we delete all the transitions or tune down the laser field. This suggests that HHG is potentially a powerful tool to detect band transitions in the crystal momentum space. Such a detection scheme is achieved by three crucial elements in HHG: (i) the incident photon energy pre-selects dipole-allowed band states, (ii) when the HHG is dispersed into the $k$ space, it further limits them to fewer band states, and (iii) HHG must result from real transitions among band states. Experimentally one must first examine the peak structure. This ensures high accuracy of our proposed scheme.

After we solve the Liouville Eq. (4), the optical HHG is not the only one that we can investigate. We can also compute the spin change through $\mathbf{S}(t) = \sum_{\mathbf{k}} \mathrm{Tr}\left[\rho_{\mathbf{k}}(t)\widehat{\mathbf{S}}_{\mathbf{k}}\right]$, where $\widehat{\mathbf{S}}_k$ is the spin operator, and then we Fourier-transform it into the frequency domain. Figure 4c shows our results. Its zeroth order is the baseline of spin moment, and reflects how much the laser pulse demagnetizes our spin system. Demagnetization is only part of the entire process, and spin also oscillates with time. Our spectrum surely catches this, but the harmonic peak only appears at even orders. This is because the spin has SU(2) symmetry, and the laser field must interact with the system at least twice to affect the spin. $S_z$ is dominated by the zeroth order. We do not find a higher-order harmonic beyond the 10th order with our current laser parameters. We caution that emission from spin in general is much weaker. However, with new experimental developments[27,29], these signals should be detectable. One advantage in systems with inversion symmetry is that the dipole radiation has no even harmonic, so the emission from spin is essentially background free. Nonlinear magneto-optical investigations in ferromagnetic monolayers and thin films have a long history. Second-harmonic generation has been extensively used to probe surface magnetism[40,43]. High-harmonic generation has already been used to probe ultrafast and element-specific magnetization[44,45], and THz emission from magnetic thin films was reported[39,46]. Thus, our findings are likely to motivate further investigations in the future (see Supplementary Information).

## Methods

**Time-dependent Liouville density functional theory**. Our theoretical calculation consists of two steps. First, we solve the Kohn–Sham equation[35] to obtain the eigenvalues and eigenstates. We employ the generalized gradient approximation at the PBE level[34] as implemented in the Wien2k code. The code employs the full-potential linearlized augmented plane-wave method, where dual basis functions are used in the atomic sphere and interstitial regions, and no approximation to the sphere is made. This makes calculations very accurate. The product of the Muffin-tin radius and plane-wave cutoff is $R_{MT}K_{\max} = 7$. In the spin–orbit coupling calculation, we use a large orbital angular momentum quantum number of $L = 6$ to ensure the high accuracy of the spin matrices; and all the eigenstates up to 3.5 Ryd are computed. This is the same maximum energy used in the optical calculation where transition matrix elements are computed. We have changed the original optic code so we can store all those matrices in an unformatted form, which improves the accuracy of the HHG calculation greatly.

Next we solve the Liouville equation for density matrices in the time domain for all the $k$ points. This step is most time-consuming since we have to solve thousands of equations simultaneously. Our code is fully parallelized using the MPI architecture. Once we find the density matrices at each time step, we compute the expectation value of the momentum operator or other interesting quantities by tracing all the product of density matrices and an operator $\mathcal{O}$, i.e. $\sum_k \mathrm{Tr}(\rho_k(t)\mathcal{O})$. Here $\rho$ depends on the space group symmetry through $H_I$, for which we show one example below.

**Symmetry analysis in an orthorhombic lattice**. The symmetry group, for a nonmagnetic orthorhombic lattice as well as for a magnetic orthorhombic lattice without spin–orbit coupling, includes all eight symmetry operations, $\{O_i\}$ where $i$ runs from 1 to 8:

$$O_1 = \begin{pmatrix} -1 & 0 & 0 \\ 0 & -1 & 0 \\ 0 & 0 & -1 \end{pmatrix}, O_2 = \begin{pmatrix} -1 & 0 & 0 \\ 0 & -1 & 0 \\ 0 & 0 & 1 \end{pmatrix}, O_3 = \begin{pmatrix} 1 & 0 & 0 \\ 0 & 1 & 0 \\ 0 & 0 & -1 \end{pmatrix}, O_4 = \begin{pmatrix} 1 & 0 & 0 \\ 0 & 1 & 0 \\ 0 & 0 & 1 \end{pmatrix}$$

(6)

$$O_5 = \begin{pmatrix} -1 & 0 & 0 \\ 0 & 1 & 0 \\ 0 & 0 & -1 \end{pmatrix}, O_6 = \begin{pmatrix} -1 & 0 & 0 \\ 0 & 1 & 0 \\ 0 & 0 & 1 \end{pmatrix}, O_7 = \begin{pmatrix} 1 & 0 & 0 \\ 0 & -1 & 0 \\ 0 & 0 & -1 \end{pmatrix}, O_8 = \begin{pmatrix} 1 & 0 & 0 \\ 0 & -1 & 0 \\ 0 & 0 & 1 \end{pmatrix}.$$

(7)

In general, there are eight different density matrices, $\rho_k[(e/m)O_i\widehat{\mathbf{P}} \cdot \mathbf{A}(t)] \equiv \rho_k(O_i)$. The symmetry operation also applies to the operator $\mathcal{O}$ as $O_i\mathcal{O}$, so the symmetrized trace is $\sum_{i=1,8} \sum_k \mathrm{Tr}[\rho_k(O_i)(O_i\mathcal{O})]$. However, for a spin-polarized and spin–orbit coupled system, only the first four operations ($O_1$, ..., $O_4$) remain in the group (Eq. (6)), while Eq. (7) is left out because they change the spin direction (Eq. (1)). This is the origin of the magneto-high-harmonic generation. For other systems, one can use the same method to work out the details.

**Data availability**. The data that support the plots within this paper and other findings of this study are available from the corresponding author upon reasonable request.

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

## Acknowledgements

This work was supported by the U.S. Department of Energy under Contract No. DE-FG02-06ER46304 (GPZ, MM, and YHB). M.S.S. acknowledges the support of the NSFC of China Under Nos. 51372107 and 11774139. Part of the work was done on Indiana State University's high performance quantum and obsidian clusters. The research used resources of the National Energy Research Scientific Computing Center, which is supported by the Office of Science of the U.S. Department of Energy under Contract No. DE-AC02-05CH11231.

## Author contributions

G.P.Z. and M.S.S. designed and carried out the calculation. G.P.Z. drafted the paper with contributions from all the authors. All the authors discussed the results. Y.H.B. parallelized the code and ported it to NERSC.

## Additional information

**Competing interests:** The authors declare no competing interests.

