## [Peer Review File · Nature Communications]

Reviewers' comments:

Reviewer #1 (Remarks to the Author):

The reviewed manuscript reports numerical simulations of high-harmonics emission by magnetic mono- and tri-layers of iron atoms, driven by intense visible (620 nm, 0.11 TW/cm²) light. Both linear (polarized perpendicular to the layers) and circular (polarized within the plane of the layers) driving light is considered. The light is treated in the velocity-gauge dipole approximation. The monolayers are taken as the (100) and (110) faces of the body-centered cubic structure.

In order to understand the results and evaluate the claims made in the manuscript it is helpful to first establish some symmetry properties of the this model system.

Let's consider a rotation by 180 degree around the z axis, $C_{\{2z\}}$. This symmetry operation transforms atomic positions to themselves, and leaves the spin labels unchanged. It changes the sign of the X and Y components of all polar and axial vectors.

The field-free Hamiltonian, including the spin-orbit terms, is left unchanged by $C_{\{2z\}}$. Furthermore, field-free states with magnetization direction along Z are left unchanged by it, possibly up to an immaterial overall phase factor.

We note that due to the way the initial states were constructed in the present work, the initial magnetization is always along the spin quantization axis used in Eq. 1, which is the Z axis.

We also notice that $C_{\{2z\}}$ leaves vector-potential linearly polarized along space Z direction unchanged. Therefore, the light interaction Hamiltonian also remains unchanged.

Thus, we must conclude that for the light linearly-polarized along the Z direction, both the full time-dependent Hamiltonian and the initial conditions are invariant with respect to $C_{\{2z\}}$. In this case, all observables, including the induced dipole, must also remain invariant to the rotation. Because the rotation changes the sign of the X and Y components of the induced dipole, we have to conclude that the induced dipole vanishes along these two directions.

In the absence of spin-orbit coupling, the same arguments also apply separately to the spin-alpha and spin-beta contributions to the induced dipole and harmonic emission.

Looking at the results presented in Fig 2, we clearly see a symmetry-breaking signal along X and Y directions. It is at the 10^0 level for the fundamental (panels a and b), dropping to 10^{-2} - 10^{-1} level for third and fifth harmonics. Any calculated harmonics, or differences between harmonics at this or lower level have to be considered to be unconverged.

Therefore, we must discard all results in the section "Spin-polarized HHG" (pages 4-5) and in Figure 2. We also must discard the discussion in the section "Crystal-momentum resolved HHG" (pages 6-7) and Figure 4: the harmonic "signal" discussed and analyzed in this part is a numerical artifact.

The results presented in section "Helicity and surface orientation dependence" and in Figure 3 do not violate symmetry constraints, and may represent a real physical effect. At the same time, the differences between the " σ^+ " and " σ^- " polarizations of the driving circular field appears to be below the 10^{-1} artifact level - so it is hard to assign any significance to these results.

To summarize, the reviewed manuscript presents numerically unconverged simulations. No useful conclusions could be drawn from these results, making the manuscript unpublishable.

Reviewer #2 (Remarks to the Author):

Please see attachment.

Reviewer #3 (Remarks to the Author):

This is a theoretical analysis of HHG in magnetic materials done with DFT to calculate eigenstates, -energies, and dipole moments; time evolution is performed by solving the time dependent Liouville equation.

1. I assume the td Liouville equation refers to the one-body density operator; please clarify.
2. Overall it is an interesting work. I have some misgivings about the signal strength; usually HHG is measured along laser polarization; in monolayers the signal is already weak enough. Here important effects are identified in the plane perpendicular to laser polarization. If I read the graphs (2) correctly, the signal in the perpendicular plane is 5-7 orders of magnitude weaker than along the pump polarization. As much of the impact of the paper rests on these perpendicular

effects, convincing arguments that these signals actually can be measured would be very helpful.

3. In general the paper is phenomenological and is lacking explanations of the identified effects. I was missing a discussion where the HHG signal comes from; perturbative / non-perturbative; interband / intraband ? The mechanism resulting in HHG with circularly polarized light is not clear?

4. Minor point: I did not find the bandstructure; as the authors talk about retrieving the bandstructure and discuss specific bandstructure related effects, a figure with the bandstructure would be helpful.

5. Retrieval of the transitions/bandstructure relies heavily on advance knowledge of the structure / transitions / bands; if one has to know all of this beforehand, can one really claim the "retrieval of bandstructure" from harmonic spectra?

In conclusion, the paper could benefit a lot from a more detailed discussion along the above points.

Reviewer #2 (Remarks to the Author):

The manuscript by Zhang *et al.* presents a novel theoretical investigation on the possibility of generating high-order harmonics from the bulk of Fe, a ferromagnetic material. They predicted that HHG can be seen for monolayer and trilayers. They used a method which they called TDLDF (Ref. 34) to investigate spin-polarized HHG and furthermore orientation dependence of HHG, k-resolved HHG, and magneto-optical HHG. To the best of my knowledge, this work indeed is the first to report theoretical investigation of HHG from Fe, a ferromagnetic material. Therefore, this manuscript is an interesting contribution to the lively new field HHG from solids, an active sub-field of attosecond science. The manuscript contains a lot of work, most of them are very interesting and useful. I would support publication if the authors could satisfy the following major and minor comments/questions.

Major comment:

Although I believe that the authors have competently carried out their research, the real impact of their predictions can hardly be judged at this very moment. It will only be revealed in the end, possibly in the years to come. In order to help the authors to make sure that they have contributed a work of real impact, I would recommend the following: the authors could use their formalisms and exact calculation to apply to MoS₂ (Ref. 20) whose HHG has been measured and reported. To facilitate a “fair” comparison of the calculated HHG spectra from two materials (Fe and MoS₂), the authors should use the maximum electric field strength allowed for each materials (known through previous experimental studies). Now, neglecting spin-polarized HHG, if the calculated HHG from Fe is at the same or few order of magnitude lower than those from MoS₂, then the predictions might make sense. Otherwise, they would not be of realistic importance. The reason why I have to emphasize this is because the maximum electric field one can impose on ferromagnetic materials is much weaker than compared to semiconductors or dielectrics, which the authors know very well. Expectedly, the generated high harmonics may be *much* weaker, taking into account how nonlinear the process is.

Minor comments/questions:

- The discussion on the theoretical formalism in both the main text and the supplementary materials is too generic. If the authors do not want to use the precious space of the main text, they should add enough, detailed information in the supplementary materials. This is a key point in allowing reproducibility of their results. And it is an important point that helps boosting the value of their work. I understand that the authors have described them in a more detailed fashion in Ref. 34, nevertheless, I found the discussion there is still not detailed enough. And the authors may not use exactly the same procedure and parameters for the work reported in Ref. 34 and this manuscript. They used GGA in Ref. 34, do they use the same here? What kind of parametrization they used in this manuscript? etc.
- Convergence: this is a crucial step in a numerical calculation. I understand that the authors are somehow limited by the computing resources available, perhaps the authors can find convincing arguments for why “our main conclusions are not affected by this convergence”?
- The authors have tried to be comprehensive in describing state-of-the-art literature. However, they presented an introduction with mixed literature. In this way, unfamiliar

readers will be confused among different classes of HHG (from gases, surface-plasmas, solids) and between theoretical and experimental works. For example, attosecond physics were built basing on mostly HHG from gases. Yet the authors almost ignored HHG from gases and focus on a variety of works of HHG without any classification. Furthermore, the authors included a theoretical work (Ref. 21) in between all experimental works. A major reformulation of the introduction paragraph should be needed. Ref. 2 and Krausz, Ivanov, RevModPhys 81, (2009) should help here.

- All the HHG spectra reported in this manuscript and supplementary materials have a smooth, broad, fast decaying background that can span ~ 3 orders of magnitude. In my opinion this is an artifact rising from “direct” Fourier transform of the $P(t)$. Because it was not mentioned, thus I guess the authors did not employ any dephasing mechanism in this work. Therefore, the calculated $P(t)$ would be ringing until the end of the time grid. Consequently, direct Fourier transform of $P(t)$ as in Eq. 3 will no doubt have to include a sharp cut at the end points. This evidently makes a broad background in the frequency domain, which the authors showed. The authors can get rid of it by using a filter (for example a hyper Gaussian) to effectively damp the two ends of the $P(t)$ before doing the Fourier transform. Once this is done, possibly the authors would be able to see few more harmonics. If this is the case, then I have to question the reliability of the whole calculations? Furthermore, in Eq. 3, it is a matter of taste, but most physicists would use a minus in front of $\Omega * t$, not a plus?
- Do the authors have a physical, intuitive answer for the change of spin-up and spin-down HHG spectra besides the fact that they got them from the simulations?
- The authors showed DOS but the energy range included in the DOS, Fig. 3, does not span to the maximum HHG energy (about 11 order, 22 eV). What would you comment on this?
- The k-resolved HHG study is interesting. However, it was not absolutely clear to me what the authors meant by “We disperse the harmonic signal in the crystal momentum space”. Does this mean that the authors calculated $P(t,k)$ before the final k-integration and showed here the corresponding $S(\omega,k)$? Perhaps the authors can elaborate on this. If what I thought is what the authors did, then this separation completely destroys the “interference” effect, an important feature of a quantum calculation. Thus, any conclusion reached would likely be not realistic.
- The magneto-optical HHG discussion is also very interesting. How is the vectorial $P(t)$ in this case? Can the author plot it? And possibly a typical $P(t)$ retrieved from their calculations?
- The authors possibly have enough calculated data at different electric field strengths. Could the authors generate a power scaling plot from these data? These would clearly clarify these harmonics are of perturbative or non-perturbative nature. Same question applies for the cut-off photon energy.
- Could the authors show a simple bandstructure (from Wien2k) of Fe so that the readers could have an idea of the bands they took and denoted on the figures in the supplementary materials?
- In Fig. 1, please add descriptions for the detector and possibly an orientation measurement the authors meant by the cone and the curve on the right. The emitted radiation would very

much be in a form of multiple, narrow bursts. Thus it would be more scientifically correct to draw it better than the rainbow CW.

- The authors used $Vfs/\text{Angstrom}$ as a unit which is not a common unit. Please use only $V/\text{Angstrom}$ as commonly used in literature.
- Please specify if one set of parameters of the electric field is used for all calculations (except few on the supplementary materials) or not.
- The spectra reported in Ref. 19, in my opinion, are not “noisy”. They are actually the double precision plus the background of the radiation which may have some amplitude above the double precision, thus, making some bumps on the double precision “noise”. Ref. 19 is a very nice work. I would recommend the authors to remove this comment and check my comment/suggestion above.
- It is known that the semiconductor Bloch equations approach is very suitable for investigating HHG from solids. This is evident through Ref. 25, 26, 27, and related references. Could the authors comment on the advantages/disadvantages comparing their methodology and the one promoted by Koch, Kira, Huttner, which is well respected?

Tran Trung Luu, to support transparent peer-review.

Reply to the Reviewers

G. P. Zhang, M. S. Si, M. Murakami, Y. H. Bai and Thomas F. George

We would like to thank all three reviewers for their helpful comments and suggestions. We have greatly benefited from those reports and have revised both our paper and Supplementary Materials extensively. These revisions are highlighted in blue in the paper and Supplementary Materials. We have carried out additional calculations to address various issues raised by the reviewers, which are both important and useful to our paper. Please see the “List of changes” for details. Below, we respond to the comments of the reviewers in their order as presented. His/her original comments start with **Reviewer**, and our response starts with the word “Reply.”

Reply to Reviewer 1

1. **Reviewer – 1** The reviewed manuscript reports numerical simulations of high-harmonics emission by magnetic mono- and tri-layers of iron atoms, driven by intense visible (620 nm, 0.11 TW/cm²) light. Both linear (polarized perpendicular to the layers) and circular (polarized within the plane of the layers) driving light is considered. The light is treated in the velocity-gauge dipole approximation. The monolayers are taken as the (100) and (110) faces of the body-centered cubic structure.

In order to understand the results and evaluate the claims made in the manuscript it is helpful to first establish some symmetry properties of the this model system.

Let’s consider a rotation by 180 degree around the z axis, C_{2z} . This symmetry operation transforms atomic positions to themselves, and leaves the spin labels unchanged. It changes the sign of the X and Y components of all polar and axial vectors.

The field-free Hamiltonian, including the spin-orbit terms, is left unchanged by C_{2z} . Furthermore, field-free states with magnetization direction along Z are left unchanged by it, possibly up to an immaterial overall phase factor.

We note that due to the way the initial states were constructed in the present work, the initial magnetization is always along the spin quantization axis used in Eq. 1, which is the Z axis.

We also notice that C_{2z} leaves vector-potential linearly polarized along space Z direction unchanged. Therefore, the light interaction Hamiltonian also remains unchanged.

Thus, we must conclude that for the light linearly-polarized along the Z direction, both the full time-dependent Hamiltonian and the initial conditions are invariant with respect to C_{2z} . In this case, all observables, including the induced dipole, must also remain invariant to the rotation. Because the rotation changes the sign of the X and Y components of the induced dipole, we have to conclude that the induced dipole vanishes along these two directions.

In the absence of spin-orbit coupling, the same arguments also apply separately to the spin-alpha and spin-beta contributions to the induced dipole and harmonic emission.

Looking at the results presented in Fig 2, we clearly see a symmetry-breaking signal along X and Y directions. It is at the 10^0 level for the fundamental (panels a and b), dropping to $10^{-2} - 10^{-1}$ level for third and fifth harmonics. Any calculated harmonics, or differences between harmonics at this or lower level have to be considered to be unconverged.

Therefore, we must discard all results in the section "Spin-polarized HHG" (pages 4-5) and in Figure 2. We also must discard the discussion in the section "Crystal-momentum resolved HHG" (pages 6-7) and Figure 4: the harmonic "signal" discussed and analyzed in this part is a numerical artifact.

Reply – We truly appreciate the reviewer’s insightful comments that allow us to correct a mistake in our original calculation. The reviewer is correct that due to the symmetry, the transverse component should vanish if the laser is polarized along the z axis. We are able to trace the source of this error to the unsymmetrized matrix elements provided by the Wien2k code. For this reason, we have now recomputed all the results and revised all the figures that are affected in our paper. This includes Figs. 2, 3 and 4 in the paper and Figs. 2, 3, 4, 6 and 7 in the Supplementary Materials. Our new results indeed verify all the reviewer’s remarks. If the laser is polarized along the z axis, there is no signal along the x and y axes, regardless of whether the system is nonmagnetic or magnetic without spin-orbit coupling. If the polarization is along the y axis, there is no signal along the x axis for nonmagnetic materials and magnetic materials without spin-orbit coupling. But there is a signal along the x axis for magnetic materials with spin-orbit coupling. This latter case, with the laser polarization along the y axis, replaces the old Fig. 2. Please see the "List of changes" for details.

2. **Reviewer – 1** The results presented in section "Helicity and surface orientation dependence" and in Figure 3 do not violate symmetry constraints, and may represent a real physical effect. At the same time, the differences between the " σ^+ " and " σ^- " polarizations of the driving circular field appears to be below the 10^{-1} artifact level - so it is hard to assign any significance to these results.

Reply – We would like to thank the reviewer for the comment. The difference between the " σ^+ " and " σ^- " polarizations is physical. Our previous incorrect results along the x and y axes in the old Fig. 2 are due to the treatment error in the Wien2k code as discussed above, not due to a numerical artifact. Our numerical noisy level is much lower than 10^{-1} . We can verify this using different laser parameters. In fact, the difference between " σ^+ " and " σ^- " is well known even for the linear magneto-optics. The reason for this difference is because different laser helicities choose a different set of states and they produce different signals. Therefore, our current result is consistent with this understanding.

3. **Reviewer – 1** To summarize, the reviewed manuscript presents numerically unconverged simulations. No useful conclusions could be drawn from these results, making the manuscript unpublishable.

Reply – Once again, we would like to thank the reviewer for the insightful comments that help us correct the error in our original paper. We have now revised the paper extensively, and we have recalculated all the results and revised all the figures that are affected in the paper.

To amplify the reviewer’s message, in the revised paper we take advantage of this opportunity to add a new subsection “Symmetry properties of magneto-high-order harmonic generation,” where we provide a real example to show how symmetry breaking by the spin-orbit coupling introduces new signals. This nicely connects with the traditional magneto-optics. In “Methods”, we have now added additional details on the symmetry properties, which should benefit the wide readership. HHG in magnetic systems has never explored before, and our study represents the first effort in this direction.

Reply to Reviewer 2

1. **Reviewer – 2** The manuscript by Zhang et al. presents a novel theoretical investigation on the possibility of generating high-order harmonics from the bulk of Fe, a ferromagnetic material. They predicted that HHG can be seen for monolayer and trilayers. They used a method which they called TDLDFT (Ref. 34) to investigate spin-polarized HHG and furthermore orientation dependence of HHG, k-resolved HHG, and magneto-optical HHG. To the best of my knowledge, this work indeed is the first to report theoretical investigation of HHG from Fe, a ferromagnetic material. Therefore, this manuscript is an interesting contribution to the lively new field HHG from solids, an active sub-field of attosecond science. The manuscript contains a lot of work, most of them are very interesting and useful. I would Support publication if the authors could satisfy the following major and minor comments/questions.

Reply – We greatly appreciate the remarks by the reviewer, and we agree with the comments.

2. **Reviewer – 2** Major comment: Although I believe that the authors have competently carried out their research, the real impact of their predictions can hardly be judged at this very moment. It will only be revealed in the end, possibly in the years to come. In order to help the authors to make sure that they have contributed a work of real impact, I would recommend the following: the authors could use their formalisms and exact calculation to apply to MoS2 (Ref. 20) whose HHG has been measured and reported. To facilitate a fair comparison of the calculated HHG spectra from two materials (Fe and MoS2), the authors should use the maximum electric field strength allowed for each materials (known through previous experimental studies). Now, neglecting spin-polarized HHG, if the calculated HHG from Fe is at the same or few order of magnitude lower than those from MoS2, then the predictions might make sense. Otherwise, they would not be of realistic importance. The reason why I have to emphasize this is because the maximum electric field one can impose on ferromagnetic materials is much weaker than compared to semiconductors or dielectrics, which the authors know very well. Expectedly, the generated high harmonics may be much weaker, taking into account how nonlinear the process is.

Reply – We would like to thank the reviewer for this excellent comment. While we are not in the best position to make a quantitative comparison between our systems and MoS2, partly because we are unfamiliar with MoS2 and we can not address this issue adequately within the limited time, our preliminary comparison demonstrates that the overall harmonic signals are comparable to each other (see

Figure 1: Preliminary comparison between a Fe(110) monolayer and MoS₂ monolayer under the same laser parameters in our paper. The black curve is for Fe(110) and the red one is for MoS₂.

Fig. 1 of this Reply). Using the maximum strength is not an option since it would easily exceed our computing time limit on NERSC (National Energy Research and Scientific Computing Center). So we carried out a calculation with the same laser parameters as in our paper. MoS₂, due to its lower symmetry and lacking of inversion symmetry, has a stronger signal in low harmonics and even-order harmonics, but at the 5th harmonic, Fe(110) has a stronger signal and has the baseline well above MoS₂. We plan to continue to work on this interesting idea in the future.

3. **Reviewer – 2 Minor comments/questions:** - The discussion on the theoretical formalism in both the main text and the Supplementary Materials is too generic. If the authors do not want to use the precious space of the main text, they should add enough, detailed information in the Supplementary Materials. This is a key point in allowing reproducibility of their results. And it is an important point that helps boosting the value of their work. I understand that the authors have described them in a more detailed fashion in Ref. 34, nevertheless, I found the discussion there is still not detailed enough. And the authors may not use exactly the same procedure and parameters for the work reported in Ref. 34 and this manuscript. They used GGA in Ref. 34, do they use the same here? What kind of parametrization they used in this manuscript? etc.

Reply – We thank the reviewer for the great suggestion. In the revised paper, we have now provided additional details by adding a section on “Method”. We choose

$RK_{max} = 7$, the orbital angular momentum quantum number of 6 for the spin-orbit coupling calculation, and the energy cutoff in both spin-orbit coupling calculations and optics calculation is 3.5 Ryd. In addition, a symmetry analysis is also added. Indeed, we use the same formalism as our prior paper Ref. 38 (Ref. 34 in the prior version). We use the same GGA. In the revised paper, we have now added two new references (Refs. 35 and 36). Please see the list of changes for details.

4. **Reviewer – 2 - Convergence:** this is a crucial step in a numerical calculation. I understand that the authors are somehow limited by the computing resources available, perhaps the authors can find convincing arguments for why “our main conclusions are not affected by this convergence”?

Reply – We appreciate the excellent question. One of our main conclusions is that HHG carries spin information. Numerically, we compare results between two different k meshes ($30 \times 30 \times 3$ and $40 \times 40 \times 4$) and find that their main difference in the momentum expectation value P is within 10%. This 10% difference is unlikely to affect our main conclusion. In the revised Supplementary Materials, we have added the similar information (see page 2 of the Supplementary Materials). Please see the “List of changes” for details.

5. **Reviewer – 2 -** The authors have tried to be comprehensive in describing state-of-the-art literature. However, they presented an introduction with mixed literature. In this way, unfamiliar readers will be confused among different classes of HHG (from gases, surface-plasmas, solids) and between theoretical and experimental works. For example, attosecond physics were build basing on mostly HHG from gases. Yet the authors almost ignored HHG from gases and focus on a variety of works of HHG without any classification. Furthermore, the authors included a theoretical work (Ref. 21) in between all experimental works. A major reformulation of the introduction paragraph should be needed. Ref. 2 and Krausz, Ivanov, RevModPhys 81, (2009) should help here.

Reply – The reviewer raises an interesting point. Due to the space limit of the paper, we regret that we can not cite many excellent references on gaseous atoms and small molecules. Instead, in this paper, we focus on those references in solids. Although Ref. 23 (Ref. 21 in the prior version) is a theoretical study based on a model, it is cited because the authors studied topological insulators, which is different from other studies. Fortunately, there are new comprehensive references available. In the revised paper, we have cited F. Krausz and M. Ivanov, Rev. Mod. Phys. **81**, 163 (2009) and S. Y. Kruchinin, F. Krausz and V. S. Yakovlev, Rev. Mod. Phys. **90**, 021002 (2018).

6. **Reviewer – 2 -** All the HHG spectra reported in this manuscript and Supplementary Materials have a smooth, broad, fast decaying background that can span 3 orders of magnitude. In my opinion this is an artifact rising from “direct” Fourier transform of the $P(t)$. Because it was not mentioned, thus I guess the authors did not employ any dephasing mechanism in this work. Therefore, the calculated $P(t)$ would be ringing until the end of the time grid. Consequently, direct Fourier transform of $P(t)$ as in Eq. 3 will no doubt have to include a sharp cut at the end points. This evidently makes a broad background in the frequency domain, which the authors

showed. The authors can get rid of it by using a filter (for example a hyper Gaussian) to effectively damp the two ends of the $P(t)$ before doing the Fourier transform. Once this is done, possibly the authors would be able to see few more harmonics. If this is the case, then I have to question the reliability of the whole calculations? Furthermore, in Eq. 3, it is a matter of taste, but most physicists would use a minus in front of $\Omega * t$, not a plus?

Reply – We appreciate the reviewer for the comment. While we are not experts in solid state HHG, we have some experience in FFT from our prior HHG studies in C_{60} and atoms (see, for instance, G. P. Zhang, Optical high harmonic generations in C_{60} , Phys. Rev. Lett. **95**, 047401 (2005) and M. Murakami, G. P. Zhang, and S.-I. Chu, Phys. Rev. A **95**, 053419 (2017)). The scenario that the reviewer mentioned did not happen in our case. Our $P(t)$ does not show a strong ringing. This is because we employ a Gaussian pulse with a very long run time, and because in metals the dephasing (which is always included in our case) is extremely fast. We followed the reviewer’s idea, but it did not help. Instead, it hides more peaks. The reason is simple, and it is well known that FFT relies on the intense sampling of data points, and the larger the number of data points is, the better the resolution becomes. Therefore, our results are extremely accurate and reliable. Here we provide one example in Fig. 2 of this Reply which shows that $P(t)$ has no strong ringing. To perform FFT, we always choose the total time length in multiples of the laser period, which ensures the highest possible accuracy. The solid curve in Fig. 2(b) shows the power spectrum without an exponential decay function, where we can see all the harmonics up to the 7th order. The red dashed curve is the one with an exponential decay from 0 fs. The curve is shifted above by 2 to see the difference. It is very clear that such a decaying function makes things worse, where the 7th order is difficult to see. We normally use $\Omega * t$, not the negative one, for our convenience.

7. **Reviewer – 2** - Do the authors have a physical, intuitive answer for the change of spin-up and spin-down HHG spectra besides the fact that they got them from the simulations?

Reply – We would like to thank the reviewer for another great question. In ferromagnets, the majority spin (spin up) has more states below the Fermi level. That is why they are called majority spins. The strength of the harmonic signal is directly proportional to the density of states (DOS) that the laser can have access. The larger the DOS, the stronger the HHG signal. By contrast, the minority spin does not have a strong DOS around the Fermi level, so their HHG signal appears to be lower. This point is explained in our paper (in the middle of the second paragraph on page 7). Please see the “List of changes” for details.

8. **Reviewer – 2** - The authors showed DOS but the energy range included in the DOS, Fig. 3, does not span to the maximum HHG energy (about 11 order, 22 eV). What would you comment on this?

Reply – We thank the reviewer for an interesting question. However, doing so would hide the details of the DOS around the Fermi level (which are much more important than those in the high-energy window), and the entire DOS would collapse into several spikes. We hope that the reviewer appreciates the difficulty here.

Figure 2: (a) $P(t)$ as a function of time in fs. There is no strong ringing. (b) Power spectrum without (solid curve) and with (dashed curve) the decaying function.

9. **Reviewer – 2** - The k -resolved HHG study is interesting. However, it was not absolutely clear to me what the authors meant by “We disperse the harmonic signal in the crystal momentum space”. Does this mean that the authors calculated $P(t,k)$ before the final k -integration and showed here the corresponding $S(\omega,k)$? Perhaps the authors can elaborate on this. If what I thought is what the authors did, then this separation completely destroys the “interference” effect, an important feature of a quantum calculation. Thus, any conclusion reached would likely be not realistic.

Reply – The reviewer is correct that we compute $P(t)$ k -point by k -point because k is a good quantum number. However, the interference is not destroyed because the total spectrum is computed after the summation over k in the time domain, not the other way around. This summation over k ensures all the interference is retained. Mathematically, we compute $\sum_k P(k,t)$ to get $P_{total}(t)$ and then Fourier transform $P_{total}(t)$ to get $P_{total}(\omega)$, where the summation over k retains the interference among the k points.

10. **Reviewer – 2** - The magneto-optical HHG discussion is also very interesting. How is the vectorial $P(t)$ in this case? Can the author plot it? And possibly a typical $P(t)$ retrieved from their calculations?

Reply – The reviewer raises a wonderful idea. We plot a phase diagram of the x and y components of $P(t)$ in Fig. 2(d) of the paper and also in Fig. 3 of this Reply. It is very interesting that P_x and P_y have a clear phase relation between them, and the major axis of the ellipse formed by P_x and P_y tilts away from the y axis, an indication of the spin-orbit coupling. This is nicely connected with our earlier work on the time-resolved magneto-optical Kerr effect, where a similar feature is observed. We plan to investigate this phase diagram further in the near future.

Figure 3: (a) $P(t)$ as a function of time in fs. There is no strong ringing. (b) Power spectrum without (solid curve) and with (dashed curve) the decaying function.

11. **Reviewer – 2** - The authors possibly have enough calculated data at different electric field strengths. Could the authors generate a power scaling plot from these data? These would clearly clarify these harmonics are of perturbative or non-perturbative nature. Same question applies for the cut-off photon energy.

Reply – We appreciate this interesting suggestion. Unfortunately, we do not have enough data sets to allow us to generate a scaling plot since our calculation is extremely time-consuming with so many k points, much more than those in semiconductors and insulators. For a stronger electric field strength, it takes an even longer time, exceeding the computing time limit at NERSC. On the other hand, we should point out that this paper only represents a beginning. Many interesting topics, such as the perturbative versus non-perturbative process and cutoff energy, are beyond the scope of this paper, and will be explored in the future.

12. **Reviewer – 2** - Could the authors show a simple bandstructure (from Wien2k) of Fe so that the readers could have an idea of the bands they took and denoted on the figures in the Supplementary Materials?

Reply – We now present a bandstructure in the paper (see Fig. 4(b)) and also Fig. 5 of the Supplementary Materials.

13. **Reviewer – 2** - In Fig. 1, please add descriptions for the detector and possibly an orientation measurement the authors meant by the cone and the curve on the right. The emitted radiation would very much be in a form of multiple, narrow bursts. Thus it would be more scientifically correct to draw it better than the rainbow CW.

Reply – This is an excellent suggestion. In the revised paper, we have now revised the figure by changing the color of the harmonic radiation and adding a few explanatory words in the figure.

14. **Reviewer – 2** - The authors used Vfs/Angstrom as a unit which is not a common unit. Please use only V/Angstrom as commonly used in literature.

Reply – We thank the reviewer for pointing this out. In the Supplementary Materials, we provide a conversion formula (see page 1). For the reviewer’s convenience, we reproduce it here. $E_0(\text{V}/\text{\AA}) = A_0(\text{Vfs}/\text{\AA})\omega$, where E_0 is the electric field amplitude, A_0 is the laser vector potential amplitude, and ω is the laser frequency.

15. **Reviewer – 2** - Please specify if one set of parameters of the electric field is used for all calculations (except few on the Supplementary Materials) or not.

Reply – Yes, we use one set of parameters of the electric field. In revised paper, we provide this information on line 21 of page 5.

16. **Reviewer – 2** - The spectra reported in Ref. 19, in my opinion, are not “noisy”. They are actually the double precision plus the background of the radiation which may have some amplitude above the double precision, thus, making some bumps on the double precision “noise”. Ref. 19 is a very nice work. I would recommend the authors to remove this comment and check my comment/suggestion above.

Reply – We agree with the reviewer and in the revised paper, we simply remove the sentence (see page 5 of the revised paper). Please see the “List of changes” for details.

17. **Reviewer – 2** It is known that the semiconductor Bloch equations approach is very suitable for investigating HHG from solids. This is evident through Ref. 25, 26, 27, and related references. Could the authors comment on the advantages/disadvantages comparing their methodology and the one promoted by Koch, Kira, Huttner, which is well respected?

Reply – The group of Koch, Kira and Huttner is a leader in this field. Our method and their method are almost identical, so there is really almost no advantage or disadvantage in this respect. In some cases, they use the Heisenberg picture and we use the density matrix formalism, but of course they are equivalent. One difference is from materials themselves. Semiconductors do not need many bands and many k points, so there are not too many operators that one has to store, but metals need several orders of magnitude more bands and k points.

Reply to Reviewer 3

1. **Reviewer – 3** This is a theoretical analysis of HHG in magnetic materials done with DFT to calculate eigen-states, -energies, and dipole moments; time evolution is performed by solving the time dependent Liouville equation.

1. I assume the td Liouville equation refers to the one-body density operator; please clarify.

Reply – The reviewer is indeed correct that we use the one-body density operator for our calculation.

2. **Reviewer – 3** 2. Overall it is an interesting work. I have some misgivings about the signal strength; usually HHG is measured along laser polarization; in monolayers

the signal is already weak enough. Here important effects are identified in the plane perpendicular to laser polarization. If I read the graphs (2) correctly, the signal in the perpendicular plane is 5-7 orders of magnitude weaker than along the pump polarization. As much of the impact of the paper rests on these perpendicular effects, convincing arguments that these signals actually can be measured would be very helpful.

Reply – We thank the reviewer for the high marks on the quality of our paper. The reviewer raised an interesting point. In the revised paper, we now align the laser field along the monolayer plane (see the revised Fig. 2). These signals are measurable, if we compare those high-harmonic ones with the linear optical one. The high-harmonic signals do not drop a significant amount. In fact, the second-harmonic signal has been measured previously in a thin film (see Ref. 41). Naturally, this also depends on the laser field amplitude. In our calculation, to be conservative, we use a very weak laser field, but we also use a slightly stronger laser field (see Fig. 2(e)), where we find that the harmonics up to the 13th order are detectable. In the revised paper, we have added a statement on the possibility to measure these harmonic signals (see the first paragraph on page 6). Please see the “List of changes” for details.

3. **Reviewer – 3 3.** In general the paper is phenomenological and is lacking explanations of the identified effects. I was missing a discussion where the HHG signal comes from; perturbative / non-perturbative; interband / intraband ? The mechanism resulting in HHG with circularly polarized light is not clear?

Reply – We again appreciate the reviewer’s insightful comments. On page 7 under the Discussion, we discuss where the HHG signal comes from. In particular, we show that once the HHG is dispersed into the crystal momentum space, it is possible to pin down the origin of HHG signal. For instance, at the Z point, the fifth order harmonic is from transitions from those states between 8 and 9.36 eV above the Fermi level to the states at -1.87 eV below the Fermi level. Thus, these harmonics mainly come from the interband transition and are highly nonperturbative. The mechanism for the results with circularly polarized light is due to the spin polarization (due to the exchange interaction) and the spin-orbit coupling. Because of the asymmetry in the spin channel, the signals are different for different light helicities. It is the same mechanism that leads to the conventional magneto-optical effects. In the revised paper, we have added a statement on this (see the blue highlighted texts on pages 6 and 8). Please see the “List of changes” for details.

4. **Reviewer – 3 4.** Minor point: I did not find the bandstructure; as the authors talk about retrieving the bandstructure and discuss specific bandstructure related effects, a figure with the bandstructure would be helpful.

Reply – We appreciate the reviewer’s comment. We have added the bandstructure in Fig. 4(b) in the paper and also in the Supplementary Materials Fig. 5 on page 10. Please see the “List of changes” for details.

5. **Reviewer – 3 5.** Retrieval of the transitions/bandstructure relies heavily on advance knowledge of the structure / transitions / bands; if one has to know all of this

beforehand, can one really claim the "retrieval of bandstructure" from harmonic spectra?

Reply – The reviewer raises a valid point, with which we agree with. For this reason, in the paper we consider HHG as a challenging technique for bandstructure mapping (see line 1 on page 8). This important issue was also raised in this year's American Physical Society March Meeting, but there is no clear answer to it. After we received the report, we investigated this further and found some additional information. In general, if the harmonic peaks are due to the virtual excitation, the peak, if plotted on the logarithmic scale, appears like a Gaussian function and symmetric with respect to multiples of photon energy. These harmonics do not carry information about bandstructures. On the other hand, if the harmonics result from true excitations among band states, then the harmonic peaks appear asymmetric and carry the band information. We plan to investigate this issue in the near future. In the revised paper, we have added this additional information (see the second paragraph on page 8 of the paper) and the last paragraph on page 10 of the Supplementary Materials. Please see the "List of changes" for details.

6. **Reviewer – 3** In conclusion, the paper could benefit a lot from a more detailed discussion along the above points.

Reply – We greatly appreciate the reviewer's recommendation which has allowed us to improve our paper enormously.

In conclusion, we have tried to satisfy and clarify all the concerns raised by all three reviewers. We are grateful for their careful reading and excellent recommendations. We want to emphasize that studies like ours should motivate new experimental and theoretical investigations. We hope that our revised paper, which has clearly benefited from the reviewers' reports, is now acceptable for publication in Nature Communications.

List of changes

Here is a list of major changes. All the changes in the paper and supplementary materials are highlighted in blue.

1. On p3, and continued on page 4, a new section is added.
2. On p5, a statement about the noisy spectrum is removed.
3. We have now added a statement on the origin of the difference between left and right circularly polarized light (see highlighted texts in blue on pages 6 and 8).
4. On page 8, in Fig. 4(b), we have added a band structure that includes all the band states.
5. On page 9 and continued on to page 10, we have added two new sections under Methods, Time-dependent Liouville density functional theory and the symmetry analysis.
6. On page 11, Refs. 4 and 5 are added.
7. On page 13, two new references Refs. 35 and 36.
8. On page 16, Fig. 2(d), we have added a phase diagram of $P_x(t)$ versus $P_y(t)$.
9. We have revised all the four figures on pages 15-18. This includes the captions.

List of changes in the supplementary materials

1. On page 3, we have added a conversion between $V_f/\text{\AA}$ and $V/\text{\AA}$
2. On page 4 and continued on page 5, we have added a rationale why we think that the convergence with k does not affect our main conclusion.
3. On page 5, we have changed Fig. 2, including its caption.
4. On page 6, we have changed Fig. 3, including its caption. The laser and system parameters are included on the figure.
5. On page 8, we have now added the band structure.
6. On page 10, we have added Figure 5 on the band structure.
7. On page 10, we have added some further discussions on band mapping.
8. On page 13, we have revised the figure.
9. On page 14, we have revised Figure 8 and added a new set of data.

Reviewers' comments:

Reviewer #1 (Remarks to the Author):

The revised version of the manuscript represents a substantial improvement over the initially-submitted text. The reported computational results are now compatible with the symmetry properties of the system. I believe these results are interesting, and should eventually be published in some form. At the same time, I also feel that the work has been submitted for publication prematurely, before the authors have fully developed and tested their ideas. As the result, the manuscript in its present form is very phenomenological, and adds very little to our understanding of HHG in solids.

The three key findings of the work are:

1. The high harmonics generated in a spin-orbit-coupled magnetic solid exhibit circular dichroism
2. These harmonics could be used to extract information on the band structure
3. Magnetic solids will also generate (even) magnetic-dipole harmonics

The first finding is obvious given the symmetry properties of the system. It is not specific to solid targets: any chiral target (or which magnetic solid is an example) will exhibit circular dichroism in HHG. Given that the effect -must- exist, the interesting and useful questions are its magnitude, and the conditions needed for its observation. I feel that these subjects are not adequately explored by the manuscript.

The second claim is not developed to the point where it is either useful or even convincing. It is true that -if- one could resolve the high-harmonics spectrum in the K space, the harmonics can be used for an all-optical determination of (the elements of) the band structure [see the work by Vampa, Corkum, et al - authors' ref. 33]. The critical question is however -how- does one do so - the harmonic emission does not carry this information per se. Vampa and Corkum have established the mapping mapping between the frequency of the harmonic, and the K value it corresponds to, using the semi-classical analysis (which was in turn validated by numerical simulations and experiment). As far as I can tell, this analysis relies critically on two properties of their target systems (ZnO): a) it has a non-zero band-gap, so that the time carriers get created by the field is well-defined; and b) there is a very small number of bands contributing to the emission. Neither property is true for the model considered here - so it is not at all clear whether the Vampa and Corkum's analysis could be extended to the magnetic monolayers.

The third claim (emission of even harmonics through magnetic-dipole radiation) is interesting, even if not entirely unexpected on the symmetry grounds. The critical question here is whether

these harmonics will be strong enough to be observed. A quick estimation shows that the magnetic-dipole harmonics are expected to be 6 orders of magnitude weaker than their electric-dipole counterpart. Here how I get the estimation:

1. The magnetic dipole is proportional to the excess magnetization. The electric dipole is proportional to the total valence charge. The former is about 1/10-th of the latter in iron and similar materials, so that we correspondingly expect magnetic dipole to be 1/10-th of the electric dipole (atomic units).
2. The coupling of the magnetic dipole to the emitted radiation contains an additional factor of the fine-structure constant (1/137), compared to the electric dipole.
3. The emission power (for either electric or magnetic dipole) is proportional to the -square- of these factors.

Overall, I then expect magnetic-dipole emission to be weaker by a factor $((1/10)*(1/137))^2$, or about 10^{-6} . It is unlikely that emission at this level could be unambiguously detected in the presence of the much-stronger electric-dipole emission at nearby odd harmonics.

Some more technical additional points:

1. The usual vehicle for the symmetry analysis in the presence of the spin-orbit interaction are double groups. Dropping operations from the point group as the authors do is also possible, but feels awkward.
2. From Eq. 4, which does not explicitly couple different K points, it appears that the simulations neglect the intra-band contributions to the harmonics. Is this correct? If yes, what is the justification? If not, then I misunderstand what is being done. Other readers may then have the same misunderstanding - so that perhaps it is a good idea to give a more detailed (but not excessively technical, for a preference) explanation of the theory part.
3. It appears that Figure 1 has not been updated to reflect the changes to the simulation parameters since the initial version: all simulations reported presently correspond to laser propagation direction normal to the surface (and presumably to the transmission geometry). Please update.
4. As already noted by one of the other referees, "V-fs/Angstrom" for the field strength are non-standard units, which are not customarily used in the field. The conversion factor given in the SM is not adequate; please switch to either V/A or TW/cm².

To summarize: this work is potentially interesting, but has been submitted prematurely.

Reviewer #2 (Remarks to the Author):

Please see attachment.

Reviewer #3 (Remarks to the Author):

I read all the referee comments and the corresponding answers. I feel that all raised concerns were satisfactorily answered and recommend publication of the manuscript in Nat Comm as is.

Reviewer #2 (Remarks to the Author):

I have read the authors' response and the revised manuscript and the supplementary materials. Before giving the final comments, I would like to discuss the analysis of the data forwarded to me from the Editor.

Fig. 1: Significant improvement of spectral analysis using proper filter. a, Real part of the raw time dependent polarization along x-axis obtained from the authors, corresponding to the spectra shown in Fig. 1 of the authors response, is shown in dark blue line. A filter of hyper Gaussian shape is plotted as the orange line. **b,** Filtered time-dependent polarization as the product of the $\text{Re}[E_{tx}]$ and the filter. **c,** Comparison between the FFT performed using the original Pt and the filtered Pt.

As I said in the comments to the authors in the first round, the smooth, broad background in the spectra is the artefact of the direct FFT of the Pt without employing a proper filter. If a filter is applied, then most of the signal after 400 fs will be damped to zero (Fig. 1b). Therefore, integral of the Pt will be zero, no causality is violated and FFT gives a much more beautiful spectrum as shown in Fig. 1c. Now with proper filter applied, the calculated spectrum shows very nicely the double precision noise floor (at around 10^{-24}) as most calculations of high-order harmonic generation showed. Furthermore, the new

spectrum shows many more harmonics that were obscured by the problematic FFT procedure before. One more thing, the hyper Gaussian used in this case does not destroy any spectra feature. It only slightly modifies the relative amplitude of the harmonics due to the final part of time domain signal is reduced. However, the perfect spectrum can only be calculated if the time duration is extended to infinity which is impossible to calculate. And the filter procedure used in this case is very close to perfect.

Now, we look at the data shown in Fig 2 of the authors' response, that I re-calculated and show here:

Fig. 2: Significant improvement of spectral analysis using proper filter – part 2. a, Real part of the raw time dependent polarization along y-axis obtained from the authors, corresponding to the spectra

shown in Fig. 2a of the authors response, is shown in dark blue line. A filter of hyper Gaussian shape is plotted as the orange line. **b**, Zoomed-in plot. **c**, Zoomed-in, filtered time-dependent polarization as the product of the $\text{Re}[E_{ty}]$ and the filter. **d**, Comparison between the FFT performed using the original Pt and the filtered Pt.

Even when the Pt does not look like it has ringing at the end of the calculation, if one looks closer, the ringing is there at the small amplitude scale. Importantly, this is strong enough to dominate the spectral domain in the logarithmic scale. Once again, with the filter applied, many more harmonics can be seen beautifully, as shown in Fig. 2d.

Fig. 3: Significant improvement of spectral analysis using proper filter – part 3. a,b,c, Comparison between the FFT performed using the original Pt and the filtered Pt, applied for the case of the spectra shown in Fig. 2a,b,c main text.

Using the same procedure, I have been able to re-draw all spectra calculated from the time-domain polarization of the authors. Figure 3 shows an example.

To summarize:

1. The time-domain polarization calculated by the authors contained a lot of spectral features that have been unfortunately hindered by improper FFT procedure. By applying the procedure that I said, I see that they truly got HHG from their calculated Pt. The harmonic orders they actually obtained are much higher than what they saw and plotted. Thus this is an important point and this shows that the manuscript contains novel contributions. Since the FFT was not deal with correctly, it is hard to make sure that the whole calculations were done properly. Especially, it is not an easy task to make sure the time propagation accurately maintain the precision of the final spectral intensity. On the other hands, the authors indeed got HHG as a final result, thus I would be in general optimistic about their whole calculations.
2. In order to make sure all harmonics and their features are captured, the authors must redo all calculations with at least twice the time steps (new_dt should be at maximum $\text{old_dt}/2$). Then they should apply the same procedure that I described here and before. I would expect that they would easily see harmonics higher than 16 order, as evident in my Fig. 1c.
3. The authors should note clearly that they used **logarithmic** scale in all of their spectral plots. It seems to me (after doing the quick calculations) that the authors plotted $|P(\omega)|$ instead of $|P(\omega)|^2$ as the proper spectral intensity calculation should be. Please make sure that this is correctly done.
4. The authors must include the comparison of Fe(110) and MoS2 in the manuscript or supplementary materials.
5. The authors have responded almost satisfactorily to my comments. I would recommend publication after the above suggestions have been carried out by the authors. I do not need to see the revision again.

Reply to the Reviewers

G. P. Zhang, M. S. Si, M. Murakami, Y. H. Bai and Thomas F. George

We would like to again thank all three reviewers for their helpful suggestions and recommendations. We are very pleased that Reviewer 3 now recommends publication, and Reviewer 2 recommends publication once all the issues that she/he has raised are addressed. We have carried out additional calculations to iron out all the details. For this reason, we have revised both our paper and the Supplementary Materials (SM) extensively. The revisions are highlighted in red in the paper and Supplementary Materials (SM). Please see the “List of changes” for details.

Below, we respond to the comments of the reviewers in their order as presented. His/her original comments start with **Reviewer**, and our response starts with the word “**Reply**.”

Reply to Reviewer 1

1. **Reviewer – 1** The revised version of the manuscript represents a substantial improvement over the initially-submitted text. The reported computational results are now compatible with the symmetry properties of the system. I believe these results are interesting, and should eventually be published in some form. At the same time, I also feel that the work has been submitted for publication prematurely, before the authors have fully developed and tested their ideas.

Reply – We agree with the reviewer that our results are interesting and should be published. While we appreciate the reviewer’s comments, we beg to differ as we believe that our study represents a fresh new advancement in investigation of high-harmonic generation (HHG) by opening an unexplored frontier in magnetic materials. Our work is mature and should be published in Nature Communications, consistent with the recommendation by Reviewers 2 and 3. Let us why this is so below.

To start with, we should mention that we have spent over 5 million computing hours at Berkeley National Laboratory’s National Energy Research Computing Center (NERSC) for the last two years to systematically develop the theory and test our ideas. Here are some examples to show how we have systematically developed the theory, tested our ideas, and explained our results. This represents one of the most challenging efforts in HHG.

- (a) We fully tested our ideas for both ferromagnetic monolayers and trilayers with different system parameters (spatial orientations, different numbers of k points, and vacuum layer thickness), and different laser parameters [laser photon energies, laser field amplitudes, pulse durations and laser helicity (linear along the x , y and z axes and circularly polarized light)] in the main text and the SM. In addition, we make a comparison with MoS₂ (see below). To our knowledge, such an extensive calculation has never been carried out before in HHG.
- (b) In order to establish our key findings, we have systematically tested our idea in a nonmagnetic monolayer, magnetic but without spin-orbit coupling monolayer, and magnetic with spin-orbit coupling monolayer. We show unambiguously that spin has important effects on how the harmonics are generated in ferromagnetic materials.

- (c) We followed the advice from Reviewer 1, have added a symmetry analysis and recalculated all the results to further improve our manuscript significantly. As pointed out by the reviewer, our results are now consistent with the symmetry analysis. For this reason, we are truly grateful to the reviewer.
- (d) Based on the suggestion from Reviewer 2 in the first round, we further computed HHG signals from MoS₂ and compared them with our ferromagnetic systems to put our results on a quantitative footing. These results are in the Supplementary Materials. Please see the “List of changes” for details.
- (e) During the second round of review, upon request of Reviewer 2, we sent our original data to the reviewer and he/she also confirmed our findings. We now use his/her method and the results are even better than those in our original manuscript. Please see the “List of changes” for details.
- (f) We followed the suggestions by both Reviewers 2 and 3 that we now include the band structure in both the main text and the SM.
- (g) We followed another suggestion from Reviewer 3 to examine the condition where the band structure mapping is possible. We find that, in general, it is very challenging to use HHG to map the band structure, but if the real transitions among band states are involved, such a band structure mapping is possible.

For these reasons, we believe that we have fully developed our theory and tested our ideas, and the results in our manuscript are mature and should be published in Nature Communications. To the best of our knowledge, there has been no comparable investigation done for HHG. The fact that both Reviewers 2 and 3 recommend publication also testifies to the quality of our work.

2. **Reviewer – 1** As the result, the manuscript in its present form is very phenomenological, and adds very little to our understanding of HHG in solids.

Reply – With great respect, we note that our manuscript is not phenomenological. It is based on the first-principles density functional theory and Liouville equation, where there are no phenomenological parameters. All the band structure and electronic and magnetic properties are computed within the density functional theory. Our work surely adds new understanding of HHG in solids beyond the existing knowledge in nonmagnetic systems. Our findings should motivate further research in this area. In the following, we highlight how our work increases our knowledge and understanding beyond nonmagnetic materials.

- (a) Before our work, there has been no prior high-harmonic generation study in ferromagnetic systems of any kind, and all the previous HHG investigations have dealt with nonmagnetic materials. Our study introduces and investigates HHG in ferromagnetic materials. We show that the HHG signals in ferromagnets carry the spin information. This should open the door to other researchers in the magnetism community. As discussed in the paper, magnetic materials are already of great importance to modern magnetic storage devices. Our findings provide a new tool to ongoing research in spintronics and femtomagnetism.

- (b) To this end, it has been unknown whether spin polarization has any effect on high-harmonic generation signals. We demonstrate that the majority spin has a stronger harmonic signal than the minority spins. Magnetic high harmonic generation has precise information of spatial orientation of the magnetic monolayers. The same material, if it is cut differently, can have different MHHG signals. This difference comes from different density of states in each spin channel.
 - (c) Although it is still very challenging, MHHG is potentially a band mapping tool. For regular HHG whose signal is from nonresonant excitation without involving the band states, the band mapping is difficult. However, if HHG results from some real transitions among band states, then they carry band information. We show one example where we can pin down the origin of harmonic peaks and assign them to specific band transitions. This paves the way to future band mapping.
 - (d) We show that different from optical HHG, magnetic high-harmonic orders are always even, due to the spin $SU(2)$ symmetry. Physically, the zeroth order of the spin moment measures the demagnetization, while the higher-order ones represent spin oscillations. This nicely presents a complete picture of how spin dynamics evolve in the frequency domain from zeroth order to higher orders. Such information is useful to laser-induced ultrafast demagnetization and spin transport. For instance, one may pump magnetic front layers and probe the emission in nonmagnetic back layers. Such a geometry has been already employed for second-order harmonics experimentally (see Ref. [47]).
3. **Reviewer – 1** The three key findings of the work are:
1. The high harmonics generated in a spin-orbit-coupled magnetic solid exhibit circular dichroism
 2. These harmonics could be used to extract information on the band structure
 3. Magnetic solids will also generate (even) magnetic-dipole harmonics

Reviewer – 1 The first finding is obvious given the symmetry properties of the system. It is not specific to solid targets: any chiral target (or which magnetic solid is an example) will exhibit circular dichroism in HHG. Given that the effect -must- exist, the interesting and useful questions are its magnitude, and the conditions needed for its observation. I feel that these subjects are not adequately explored by the manuscript.

Reply – We appreciate the reviewer’s providing this remark. It is indeed true that our prediction is not limited to solid targets. Although our findings might appear obvious, no one has ever proposed HHG in a magnetic material, and our results are interesting and important. We believe that we have adequately investigated both the magnitude of our HHG signal (see the following items (a) and (b)) and investigated optimal conditions (see the following item (c)) for experimental testing through three carefully conceived levels of thoughts.

- (a) Both linear and nonlinear magneto-optic signals have been detected routinely for several decades (see Refs. [42] and [44] and other references cited there).

The results in Fig. 2 show that the magnitude of high-order harmonics does not drop significantly, and they only differ by 1-2 orders of magnitude, so they should be detectable. For instance, more recently, there are two experiments carried out on lower-order harmonics, (see Refs. [41] and [47]), where those authors already observed emissions from their samples. As explained in Item 5 below, the modern detection scheme can reach a sensitivity of 10^{-9} .

- (b) We followed the advice by Reviewer 2 and made an absolute comparison, under the same laser conditions, with a MoS₂ monolayer whose harmonic signals have been well documented. The signals from Fe (110) monolayers are comparable to those from the MoS₂ monolayer. A detailed comparison is provided in Section V of our SM, and in Fig. 11 on page 22 of SM.
- (c) We find the optimal conditions for experimental detection. For instance, we show that the Fe(110) surface has a stronger HHG signal than the Fe(001) surface. In addition, HHG is not limited to monolayer ferromagnets. We have carried out additional investigations in iron trilayers. These systems are much more complicated than the monolayer ones, and we find the HHG signals are also easily detectable. For the majority of our study, we employ a moderate laser field, so an experimentalist can easily test our results. When we increase the laser field amplitude slightly, the magnitude of harmonic signals grows sharply (see Fig. 2(e)) all the way up to the 19th order. This further demonstrates that our predicted signal is observable experimentally.

Through these multiple steps, we believe that we have adequately addressed both the magnitude of harmonic signals and their optimal condition. Please see the “List of changes” for details.

4. **Reviewer – 1** The second claim is not developed to the point where it is either useful or even convincing. It is true that -if- one could resolve the high-harmonics spectrum in the K space, the harmonics can be used for an all-optical determination of (the elements of) the band structure [see the work by Vampa, Corkum, et al - authors’ ref. 33]. The critical question is however -how- does one do so - the harmonic emission does not carry this information per se. Vampa and Corkum have established the mapping between the frequency of the harmonic, and the K value it corresponds to, using the semi-classical analysis (which was in turn validated by numerical simulations and experiment). As far as I can tell, this analysis relies critically on two properties of their target systems (ZnO): a) it has a non-zero band-gap, so that the time carriers get created by the field is well-defined; and b) there is a very small number of bands contributing to the emission. Neither property is true for the model considered here - so it is not at all clear whether the Vampa and Corkum’s analysis could be extended to the magnetic monolayers.

Reply – We thank the reviewer for raising this important question. It is fair to say that any new tools take time to develop. This is certainly true for HHG as the band mapping technique. We are aware of the challenge even in the beginning of our project. For this reason, in our main text, we call it a challenging detection scheme. When we replied to the third reviewer in the first round of reviewing, we discussed this issue. We quote some of the discussions here. This important issue was also raised in this year’s American Physical Society March Meeting, but there is no clear

answer to it. The Vampa-Corkum method is very useful, but is still limited to a small number of bands. We can not solely rely on it. After we received the prior report, we investigated this further and found some useful information. In general, if the harmonic peaks are due to the virtual excitation, the peak, if plotted on the logarithmic scale, appears like a Gaussian function and symmetric with respect to multiples of the photon energy (see the inset in Fig. 4(a)). These harmonics do not carry information about band structures, so the crystal momentum information can not be obtained by looking at the harmonic spectrum. On the other hand, if the harmonics result from true excitations among band states, then the harmonic peaks appear asymmetric with respect to multiples of the photon energy. These harmonics carry the band information, which potentially allows one to map the bands. We plan to investigate this issue in the near future.

5. **Reviewer – 1** The third claim (emission of even harmonics through magnetic-dipole radiation) is interesting, even if not entirely unexpected on the symmetry grounds. The critical question here is whether these harmonics will be strong enough to be observed. A quick estimation shows that the magnetic-dipole harmonics are expected to be 6 orders of magnitude weaker than their electric-dipole counterpart.

Reviewer – 1 Here how I get the estimation:

Reviewer – 1 1. The magnetic dipole is proportional to the excess magnetization. The electric dipole is proportional to the total valence charge. The former is about 1/10-th of the latter in iron and similar materials, so that we correspondingly expect magnetic dipole to be 1/10-th of the electric dipole (atomic units).

Reviewer – 1 2. The coupling of the magnetic dipole to the emitted radiation contains an additional factor of the fine-structure constant (1/137), compared to the electric dipole.

Reviewer – 1 3. The emission power (for either electric or magnetic dipole) is proportional to the -square- of these factors. Overall, I then expect magnetic-dipole emission to be weaker by a factor $((1/10)*(1/137))^2$, or about 10^{-6} . It is unlikely that emission at this level could be unambiguously detected in the presence of the much-stronger electric-dipole emission at nearby odd harmonics.

Reply – We appreciate the reviewer’s thoughtful question. It is true that the magnetic-dipole radiation is much weaker, but we argue that the signal should be detectable for the following reasons. First, for a system with inversion symmetry, the much stronger electric-dipole emission does not contribute to even harmonics. Provided that electric quadrupole radiation is much weaker than magnetic dipoles, the signal from the magnetic-dipole is essentially background free, as odd harmonics are 2 eV (our photon energy) away from the even-order harmonics. The electric quadrupole contribution can be subtracted using two different magnetic field directions, similar to magneto-optics. Second, current experimental detection systems should be able to probe 10^{-6} weaker signals. We take our data in Fig. 2(d) as an example. Consider the third harmonic signal as our reference. It has a magnitude of 10^1 . Then the second-harmonic signal from magnetic-dipole emission would be at 10^{-5} . Experimentally, Schubert et al. (Ref. [27]) used a combination of electro-optic sampling, an InGaAs diode array and a silicon CCD maps out HHG spectra down to 10^{-9} . Similarly, Hohenleutner et al. (Ref. [29]) showed in their extended data

(Fig. 1) that their monochromator with a calibrated piezoelectric detector, a lead sulfide diode and spectrometers employing InGaAs and cooled Si detectors can probe emission down to 10^{-9} . These two experiments provide strong evidence that those even harmonics from magnetic dipoles should be within reach of current detection schemes. At present we are looking forward to an actual experiment test. Theoretically, as pointed out by the reviewer, even-order harmonics from magnetic dipoles are very interesting by themselves. This combines the beauty of group theory and quantum magnetism theory and may allow one to probe magnetic properties of a magnet through HHG. Our results, once published, will motivate new experimental activities in HHG. The potential impact may be beyond our current imagination. The reviewer’s comment is important, so in our revised manuscript, we have now added a statement in the abstract on page 2, a similar discussion both on page 9 of the main text and page 18 in the caption of Fig. 4. Please see the “List of changes” for details.

6. Reviewer – 1 Some more technical additional points:

Reviewer – 1 1. The usual vehicle for the symmetry analysis in the presence of the spin-orbit interaction are double groups. Dropping operations from the point group as the authors do is also possible, but feels awkward.

Reply – The reviewer raises another interesting point. Our choice is only for an easy numerical implementation in our codes. As the reviewer knows very well, using double groups requires to construct the transformation matrices for spins, but the rotation angles for each symmetry operation are not given in the Wien2k code. So we have to figure out the actual rotational angles for each symmetry operator. In our method, since these symmetry operators are already given by the Wien2k code in the case.outputs file, it is much simpler to implement it numerically. We hope that the reviewer appreciates our challenge.

7. Reviewer – 1 2. From Eq. 4, which does not explicitly couple different K points, it appears that the simulations neglect the intra-band contributions to the harmonics. Is this correct? If yes, what is the justification? If not, then I misunderstand what is being done. Other readers may then have the same misunderstanding - so that perhaps it is a good idea to give a more detailed (but not excessively technical, for a preference) explanation of the theory part.

Reply – We would like to thank the reviewer for this excellent suggestion. It is true that our current calculation does not couple different k points. The intraband contributions are included indirectly at each k point through the interband transitions, but not between different k points. The reason why we do not include the intraband contribution is because our current laser amplitude is still too weak to see the effect of the intraband contribution, and the wavevector shift is very small. The second reason is numerical. If we couple all the k points, we are limited to only a smaller total number of k points since the k point parallelization is not possible, and our computers can not handle an enormous memory requirement. This would lead to the convergence problem with the number of k points. In the revised paper, we have now included a similar discussion on this. Please see the “List of changes” for details.

8. **Reviewer – 1 3.** It appears that Figure 1 has not been updated to reflect the changes to the simulation parameters since the initial version: all simulations reported presently correspond to laser propagation direction normal to the surface (and presumably to the transmission geometry). Please update.

Reply – We greatly appreciate the reviewer pointing this out to us. In the revised Figure 1, we have now oriented the laser pulse as normal incident. Please see the “List of changes” for details.

9. **Reviewer – 1 4.** As already noted by one of the other referees, “V-fs/Angstrom” for the field strength are non-standard units, which are not customarily used in the field. The conversion factor given in the SM is not adequate; please switch to either V/A or TW/cm².

Reply – We would like to thank the reviewer again. We have now changed all “V-fs/Angstrom” units to “V/Angstrom” in both the main text and SM.

10. **Reviewer – 1** To summarize: this work is potentially interesting, but has been submitted prematurely.

Reply – We greatly appreciate the excellent comments by the reviewer. It is fair to say that our current research only represents a beginning, but we feel that it is mature to be published. There are many interesting and challenging questions in this field. We plan to continue working on this. We hope that the reviewer appreciates our challenge.

Reply to Reviewer 2

1. **Reviewer – 2** I have read the authors’ response and the revised manuscript and the supplementary materials. Before giving the final comments, I would like to discuss the analysis of the data forwarded to me from the Editor. As I said in the comments to the authors in the first round, the smooth, broad background in the spectra is the artifact of the direct FFT of the Pt without employing a proper filter. If a filter is applied, then most of the signal after 400 fs will be damped to zero (Fig. 1b). Therefore, integral of the Pt will be zero, no causality is violated and FFT gives a much more beautiful spectrum as shown in Fig. 1c. Now with proper filter applied, the calculated spectrum shows very nicely the double precision noise floor (at around 10-24) as most calculations of high-order harmonic generation showed. Furthermore, the new spectrum shows many more harmonics that were obscured by the problematic FFT procedure before. One more thing, the hyper Gaussian used in this case does not destroy any spectra feature. It only slightly modifies the relative amplitude of the harmonics due to the final part of time domain signal is reduced. However, the perfect spectrum can only be calculated if the time duration is extended to infinity which is impossible to calculate. And the filter procedure used in this case is very close to perfect. Now, we look at the data shown in Fig 2 of the authors’ response, that I re-calculated and show here: Even when the Pt does not look like it has ringing at the end of the calculation, if one looks closer, the ringing is there at the small amplitude scale. Importantly, this is strong enough to dominate the spectral domain in the logarithmic scale. Once again, with the filter applied, many more harmonics can be seen beautifully, as shown in Fig. 2d.

Using the same procedure, I have been able to re-draw all spectra calculated from the time-domain polarization of the authors. Figure 3 shows an example.

Reply – We are extremely grateful for the reviewer’s comments. We use his/her method and we can now reproduce nearly all of his/her results. We design two window functions, hyper Gaussian \mathcal{W}_1 and hyperbolic tangent \mathcal{W}_2 ,

$$\mathcal{W}_1(t) = \exp \left[-(at)^8 \times b \right], \quad (1)$$

where t is in the unit of fs. Two constants are chosen so we have a window across the entire data set. We find $a = 0.035/\text{fs}$, $b = 5 \times 10^{-9}$ (no unit) to be very good for cutting off the tail of the momentum expectation value $P(t)$, while setting the leading edge at -400 fs and the trailing edge at 400 fs. Here,

$$\mathcal{W}_2(t) = (\tanh[(t + t_1) * b_1] - \tanh[(t + t_2) * b_2],) / 2 \quad (2)$$

where t_1 and t_2 set the centers of the cutoff in the beginning and end, respectively, and b_1 and b_2 are the respective widths of the leading edge and trailing edge. We find that \mathcal{W}_2 is more flexible since it has two separate control parameters. In the revised SM, we have added a similar description of our window functions. Please see the “List of changes” for details.

2. **Reviewer – 2** To summarize: 1. The time-domain polarization calculated by the authors contained a lot of spectral features that have been unfortunately hindered by improper FFT procedure. By applying the procedure that I said, I see that they truly got HHG from their calculated Pt. The harmonic orders they actually obtained are much higher than what they saw and plotted. Thus this is an important point and this shows that the manuscript contains novel contributions. Since the FFT was not deal with correctly, it is hard to make sure that the whole calculations were done properly. Especially, it is not an easy task to make sure the time propagation accurately maintain the precision of the final spectral intensity. On the other hands, the authors indeed got HHG as a final result, thus I would be in general optimistic about their whole calculations.

Reply – Once again, we greatly appreciate the great input. We have now recalculated all the FFT spectra using the reviewer’s method, and have accordingly revised all the figures. Our window functions and additional testing are presented in the SM. Please see the “List of changes” for details.

3. **Reviewer – 2** 2. In order to make sure all harmonics and their features are captured, the authors must redo all calculations with at least twice the time steps new dt should be at maximum old $dt/2$). Then they should apply the same procedure that I described here and before. I would expect that they would easily see harmonics higher than 16 order, as evident in my Fig. 1c.

Reply – We thank the reviewer for raising another interesting point. We redid all the calculations with a half-time step, and we find, with the exception in Figs. 2(e) and 3(a) and 3(b), that there is no high well-defined harmonic beyond 16. Figure (1) of this response shows an example where the peak at 17 is almost at the noise level.

Figure 1: HHG signal in Fe(110) monolayer with half of the time step. There is no more higher harmonic peak after 16. The one at 17 is already at the noise level.

4. Reviewer – 2 3. The authors should note clearly that they used logarithmic scale in all of their spectral plots. It seems to me (after doing the quick calculations) that the authors plotted $|P(w)|$ instead of $|P(w)|^2$ as the proper spectral intensity calculation should be. Please make sure that this is correctly done.

Reply – We thank the reviewer for the excellent note. In the revised paper, we have now clearly mentioned this. We have revised all the figures to reflect this change. We normally use $|P(\Omega)|$ since it gives more information to the reader. Please see the “List of changes” for details.

5. Reviewer – 2 4. The authors must include the comparison of Fe(110) and MoS2 in the manuscript or supplementary materials.

Reply – This is a great idea. We include the comparison in our SM. Please see the “List of changes” for details.

6. Reviewer – 2 5. The authors have responded almost satisfactorily to my comments. I would recommend publication after the above suggestions have been carried out by the authors. I do not need to see the revision again.

Reply – We truly appreciate the reviewer’s recommendation of publication of our work.

Reply to Reviewer 3

1. **Reviewer – 3** I read all the referee comments and the corresponding answers. I feel that all raised concerns were satisfactorily answered and recommend publication of the manuscript in Nat Comm as is.

Reply – We greatly appreciate the reviewer’s recommendation.

In conclusion, we are grateful to all three reviewers for their careful reading and excellent recommendations. We should emphasize again that studies like ours should motivate new experimental and theoretical investigations. We hope that our revised paper, which has clearly benefited from the reviewers’ reports, is now acceptable for publication in Nature Communications.

List of changes

Here is a list of major changes. The changes in the paper and Supplementary Materials are highlighted in red.

1. On page 2, we have inserted “though probably weak.”
2. On page 5, lines 10-12, we add the details about the intraband transition; line 14, the information about the window function and how the power spectrum is computed is added; lines 18-19, we add the details about the time step used in our calculation; lines 21-22, we change the electric field unit to $\text{V}/\text{\AA}$, and lines 27-28, the information about the window function is added.
3. On page 6, line 16, we change the electric field units to $\text{V}/\text{\AA}$.
4. On page 7, lines 11-13, we slightly change the ratio due to the new FFT procedure used.
5. On page 9, lines 15-20, we have added a caution that the radiation from the spin is much weaker.
6. On page 15, we have now changed Fig. 1, so the incident direction of the laser pulse reflects what we actually simulate in the paper.
7. On page 16, in Fig. 2, we add “Logarithmic” on both vertical axes. In the caption, we have also added “Logarithmic”. All the data in all the figures are reprocessed using the window function. The electric field unit is changed to $\text{V}/\text{\AA}$. In Fig. 2(e), we have rerun a calculation with a smaller time step, and the harmonic order is now much higher. The spectrum is also processed with a window function.
8. On page 17, all the spectra in Figs. 3(a), 3(b) and 3(c) are recalculated with the filtered FFT, and we also add word “logarithmic” in Figs. 3(a), 3(b) and 3(c) and in the caption. In addition, we have added a comment on the window function that artificially blurs the difference between the x and y of the momentum. In real time dynamics, $P_x(t)$ is smaller than $P_y(t)$ for most of the time, which is further explained in the Supplementary Materials.
9. On page 18, we have recalculated FFT for Figs. 4(a) and 4(c) using the filter function. We also add the word “logarithmic” in Fig. 4 and in the caption. In the caption, we add a caution that the signal from spin might be weak.

List of changes in the Supplementary Materials

1. On page 2, we include a statement on MoS_2 .
2. On page 3-4, we have a new paragraph on the window function for FFT.
3. On page 4-5, we have added a new section on the Fourier transform. In addition, we have added two new figures on pages 13 and 14, Figs. 2 and 3.

4. On pages 6 and 7, we change the electric field units to $V/\text{\AA}$.
5. On page 10, we change the electric field units to $V/\text{\AA}$. We also modify several sentences on the new data.
6. On page 11, we have added a new section on the comparison between the Fe(110) and MoS₂ monolayers.
7. On page 12, we have added a new reference, Ref. 3, on MoS₂.
8. On page 13, we have a new figure, Fig. 2, on the window for FFT.
9. On page 14, we have a new figure, Fig. 3, on how the window function affects the FFT spectrum.
10. On page 15, we have added “Logarithmic” along the y axes. We have redrawn all the parts of Fig. 4 using the hyper Gaussian function. In addition, we change the electric field units to $V/\text{\AA}$ in the caption.
11. On page 16, we replot all the parts of Fig. 5 using the hyper Gaussian window function. In addition, we change the electric field units to $V/\text{\AA}$ in the caption and in the figure.
12. On page 17, we replot all the parts of Fig. 6 using the the hyper Gaussian window function. In addition, we change the electric field units to $V/\text{\AA}$ in the caption.
13. On page 20, we replot all the parts of Fig. 9 using the hyper Gaussian window function.
14. On page 21, we replot all the parts of Fig. 10 using the hyper Gaussian window function. In addition, we change the electric field units to $V/\text{\AA}$ in the caption.
15. On page 22, we add a new figure, Fig. 11, to compare the harmonic signals between the Fe(110) and MoS₂ monolayers. Harmonics from MoS₂ are vertically shifted for clarity.

Reviewers' comments:

Reviewer #1 (Remarks to the Author):

I appreciate the genuine effort taken by the authors to address comments from all the reviewers, and I thank the authors for their carefully-prepared reply.

Nonetheless, I still find that the work, while undeniably technically very challenging and elaborate, does not present significant results, physical mechanisms, or experimental proposals which are unexpected or surprising given the current state of knowledge of laser-matter interactions.

Furthermore, the revised version fails to meaningfully address the key issue of how to resolve the HHG emission in this system in the K space. Without at least a plausible answer to this question, the claim of assigning harmonic peaks to specific transitions and thereby learning about the material's band structure is meaningless and misleading.

Finally, I find no fair, satisfactory discussion of the intensity and observability of calculated magnetic-dipole emission neither in the main text nor in the supplementary material.

Overall, I feel that once revised to offer a less sensationalist presentation and claims, this work will be a worthy and well-cited addition to a more specialized, technical journal. It does not belong to a general-science publication like Nature Communications, which requires a broader insight, perspective, and an advance in understanding.

In conclusion, I would like to make couple of remarks which do not affect my overall assessment of the manuscript, but which the authors may find useful:

1. The cost of the simulation by itself (see p.1 of the Reply to the Reviewers) has no bearing neither on the quality of the work nor on its significance. It is not helpful to bring it up as an argument in this context.
2. Design of Fourier window functions (pp 7-8 of the reply) is a non-trivial art, with many pitfalls for the unwary. Thankfully, there is an extensive literature on window-function properties and design; the authors may find a technical report by Heinzl et al [<http://edoc.mpg.de/395068>] to be a useful departure point.

Reply to the Reviewers

G. P. Zhang, M. S. Si, M. Murakami, Y. H. Bai and Thomas F. George

We would like to again thank the reviewers for their helpful suggestions and recommendations. In accordance with Reviewer 1 (the other two reviewers recommend publication), we have revised both our paper and the Supplementary Materials (SM), where the revisions are highlighted in red. At the end of the Reply, please see the “List of changes” for details.

Below, we respond to the comments of Reviewer 1 in their order as presented. His/her original comments start with **Reviewer**, and our response starts with the word **Reply**.

Reply to Reviewer 1

1. **Reviewer** – I appreciate the genuine effort taken by the authors to address comments from all the reviewers, and I thank the authors for their carefully-prepared reply.

Reply – We appreciate the reviewer for this nice remark.

2. **Reviewer** – I Nonetheless, I still find that the work, while undeniably technically very challenging and elaborate, does not present significant results, physical mechanisms, or experimental proposals which are unexpected or surprising given the current state of knowledge of laser-matter interactions.

Reply – It is fair to say that any research represents a small step toward a new frontier. As we discuss in the paper and responses to the reviewers, the significance of our study is that we expand the scope of high-harmonic generation (HHG) into magnetic materials (MHHG) and show how the spin degree of freedom affects how harmonics are generated. Our findings are extremely important since magnetic materials are essential to spintronics, all-optical spin switching and femtomagnetism, and are critical to information technology. MHHG presents an opportunity and potentially a new tool for researchers in these fields to investigate magnetic properties. We also understand the physical mechanism of MHHG: In magnetic materials, the majority and minority spins have different densities of states, so transitions are different for the majority and minority channels. The results that we predict are quite different and unexpected from the existing knowledge of HHG, which is solely based on nonmagnetic materials. As pointed out by Reviewer 2, there has been no prior study on magnetic materials. Our study represents the first effort that connects high-order harmonics to magnetic materials. This is the single most important contribution to the existing knowledge of laser-matter interactions.

3. **Reviewer** – Furthermore, the revised version fails to meaningfully address the key issue of how to resolve the HHG emission in this system in the K space. Without at least a plausible answer to this question, the claim of assigning harmonic peaks to specific transitions and thereby learning about the material’s band structure is meaningless and misleading.

Reply – We appreciate the reviewer for raising this important question. In the paper, we have already briefly addressed this issue from line 25 on page 8 to line 6 on page 9. In the revised SM, we have now added an entirely new section, Section VI, to extensively discuss how HHG emission can be resolved in K space. Here

is a short summary. Our results show that one should categorize harmonic peaks into two types. Harmonics in Type I are associated with virtual transitions, and harmonics in Type II are associated with real transitions. Harmonics in Type I contain contributions from multiple states through virtual transitions. Even if we disperse them into crystal momentum space (see Fig. 4 of the paper), their spectra do not change much with crystal momentum, so they can not be used as a way to disperse the band structure. Harmonics in Type II are different. They engage real transitions among band states. They can be traced back to the band dispersion, as shown in Fig. 4 of the paper. We have also changed the abstract to reflect the nature of K-resolved band structure mapping. Please see the “List of changes” for details.

4. **Reviewer – 1** Finally, I find no fair, satisfactory discussion of the intensity and observability of calculated magnetic-dipole emission neither in the main text nor in the supplementary material.

Reply – We thank the reviewer for the comment. This reminds us of our prior response to the reviewer. In the revised SM, we have now added a new section, Section VII, to discuss the intensity and observability of the calculated magnetic-dipole emission. The material is based on our original response to the reviewer. For the reviewer’s convenience, we summarize them here. We assume a similar reduction of magnetic-dipole emission with respect to electric-dipole emission. Then we compare it with the existing experimental detection efficiency, and we find that the signal should be detectable. Please see the “List of changes” for details.

5. **Reviewer – 1** Overall, I feel that once revised to offer a less sensationalist presentation and claims, this work will be a worthy and well-cited addition to a more specialized, technical journal. It does not belong to a general-science publication like Nature Communications, which requires a broader insight, perspective, and an advance in understanding.

Reply – We appreciate the reviewer’s remarks. With great respect, we disagree with the notion that our research does not belong in Nature Communications. Our research results represent a major breakthrough in magnetic high harmonic generation over the last seven years by opening an unexplored frontier. The fact that both Reviewers 2 and 3 recommend publication testifies to the quality of our work. We strongly believe our work will not only be well-cited, but will also motivate further investigations in other exotic materials. We ourselves are also regular reviewers for Nature and its sister journals. It is very important to support new ideas, since the potential impact may be beyond our current imagination. We sense that the reviewer indeed sees the significance of our research, so we would like to strongly encourage Reviewer 1 to join the other two reviewers to support our work.

6. **Reviewer – 1** In conclusion, I would like to make couple of remarks which do not affect my overall assessment of the manuscript, but which the authors may find useful:

Reviewer – 1 1. The cost of the simulation by itself (see p.1 of the Reply to the Reviewers) has no bearing neither on the quality of the work nor on its significance. It is not helpful to bring it up as an argument in this context.

Reply – We appreciate the reviewer’s advice.

Reviewer – 1 2. Design of Fourier window functions (pp 7-8 of the reply) is a non-trivial art, with many pitfalls for the unwary. Thankfully, there is an extensive literature on window-function properties and design; the authors may find a technical report by Heinzl et al [<http://edoc.mpg.de/395068>] to be a useful departure point.

Reply – We thank the reviewer for the reference. It is an interesting study, but they did not use the hyper Gaussian window function.

In conclusion, we are grateful to the reviewer for his/her careful reading and excellent suggestions. We should emphasize again that studies like ours should motivate new experimental and theoretical investigations. We hope that our revised paper, which has clearly benefited from the reviewers’ reports, is now acceptable for publication in Nature Communications.

List of changes

Here is a list of major changes. All the changes in the paper and Supplementary Materials (SM) are highlighted in red.

1. On page 2 of the paper, we have revised the sentence about “the k-resolved band structure.”
2. On page 12 of SM, we have added a new section, Section VI, on “Crystal-momentum resolved high harmonic generation.”
3. On page 12 of SM, we have added a new section, Section VII, on “Magnetic-dipole emission.”
4. On page 13 of SM, we have added three new references, Refs. 4, 5 and 6.